# The pollen virome of wild plants and its association with variation in floral traits and land use

Andrea M. Fetters[1,2✉], Paul G. Cantalupo [1,3], Na Wei [1,4], Maria Teresa Sáenz Robles[1], Aiden M. Stanley [1], Jessica D. Stephens[1,5], James M. Pipas[1] & Tia-Lynn Ashman [1✉]

Pollen is a unique vehicle for viral spread. Pollen-associated viruses hitchhike on or within pollen grains and are transported to other plants by pollinators. They are deposited on flowers and have a direct pathway into the plant and next generation via seeds. To discover the diversity of pollen-associated viruses and identify contributing landscape and floral features, we perform a species-level metagenomic survey of pollen from wild, visually asymptomatic plants, located in one of four regions in the United States of America varying in land use. We identify many known and novel pollen-associated viruses, half belonging to the Bromoviridae, Partitiviridae, and Secoviridae viral families, but many families are represented. Across the regions, species harbor more viruses when surrounded by less natural and more human-modified environments than the reverse, but we note that other region-level differences may also covary with this. When examining the novel connection between virus richness and floral traits, we find that species with multiple, bilaterally symmetric flowers and smaller, spikier pollen harbored more viruses than those with opposite traits. The association of viral diversity with floral traits highlights the need to incorporate plant-pollinator interactions as a driver of pollen-associated virus transport into the study of plant-viral interactions.

[1] Department of Biological Sciences, University of Pittsburgh, 4249 Fifth Avenue, Pittsburgh, PA 15260, USA. [2] Department of Evolution, Ecology, and Organismal Biology, The Ohio State University, 318 W. 12th Avenue, Columbus, OH 43210, USA. [3] Department of Biomedical Informatics, University of Pittsburgh, 5607 Baum Boulevard, Pittsburgh, PA 15206, USA. [4] The Holden Arboretum, 9500 Sperry Road, Kirtland, OH 44094, USA. [5] Department of Biology, Westfield State University, 577 Western Avenue, Westfield, MA 01086, USA. ✉email: fetters.38@osu.edu; tia1@pitt.edu

To reproduce, ~90% of flowering plants depend on animals, especially insects, to distribute their sperm[1], which is housed in pollen grains. Pollinating insects visit hundreds of flowers per day and may visit multiple species of plants. In so doing, they bring pollen directly to the stigma, the least defended surface of a plant[2]. Thus, viruses on or harbored within pollen grains[3–8] can be carried to new hosts by insects[9–16] and delivered to susceptible plant cells via the pollen tube, which delivers sperm to the eggs[4,17–19]. Although a growing number of studies demonstrate pollinator-mediated infection of plants with known viruses[16,20–22], and plant viruses have been found in association with pollen collected[5,20] and deposited by bees[23], no study has broadly characterized the pollen virome.

Of the nearly 1,500 viral species known to infect plants, about 75 have been shown to be associated with pollen[9–15]. Most known pollen-associated viruses are members of the Bromoviridae, Partitiviridae, and Secoviridae families. Viruses have been found both on the external surface (exine) or within the haploid (gametophytic) cells of pollen grains[3–8]. Still, our knowledge of the pollen virome as a whole is sparse and weighted toward mechanistic studies on viral infection using agricultural plant species. For example, nearly all of the pollen-associated viruses identified thus far, such as *Cucumber mosaic virus, Raspberry bushy dwarf virus, Tobacco streak virus*, and *Prunus necrotic ringspot virus*, can cause devastating damage to crops[4,9], making viral pathogens a significant challenge to global food security[24,25]. Yet the pollen virome of wild and agricultural plants remains largely uncharacterized, as does our knowledge of asymptomatic or mutualistic infections that might influence plant fitness.

Land use practices such as agricultural intensification and urbanization often fragment wild habitats, reduce native vegetation, promote invasive species establishment, and create new biotic associations in wild plant communities[26,27]. Because land use changes can also alter plant community composition, they can lead to novel plant-plant associations, including those between wild, introduced, and cultivated plants. New plant-plant interactions might increase the potential for viral spread because plant viruses can be more prevalent in areas of dense monoculture and cultivation[26–37]. Moreover, land use change can create new plant-pollinator associations as pollinators move between habitats. For example, the wide diet breadth of super generalist pollinators (e.g., the honey bee *Apis mellifera*[27]) may allow them to vector pollen-associated viruses broadly and potentially extend virus host ranges. In fact, virus-plant interactions often lie on a mutualism-antagonism continuum, and shifts in these interactions are often mediated by the environment[38,39]. A broad sampling of plant hosts in geographic regions that vary ecologically and in land use is needed to allow for a full characterization of the pollen virome.

In this work, we address fundamental gaps in the knowledge of pollen-associated plant viruses. Our first goal is to undertake a metagenomic study of the pollen virome. Our metagenomics approach allows us to capture all viruses present[40–45], including pathogenic, neutral, and possibly mutualistic ones[38], as well as to identify known viruses in hosts not previously recognized to perhaps be within their host ranges[46] and novel viruses not previously detected or described. We leverage the power of metagenomics and wide species-level sampling to characterize the pollen viromes of 24 wild, visually asymptomatic plant species (from 16 families and five subclasses), each growing in one of four geographic regions in the United States (Fig. 1). We then use phylogenetically controlled analyses to determine that: (1) pollen-associated viruses are not limited to a few, previously recognized viral families; (2) pollen-associated viruses are heterogeneously distributed across geographic regions differing in amounts of human land use and across plant subclasses; and (3) pollen-associated virus taxonomic richness correlates with floral and pollen grain traits important for plant interactions with pollinators that serve as pollen vectors.

## Results

**Plant species and sampled regions.** To broadly characterize the pollen virome in visually asymptomatic, wild plant species, we collected pollen at the species level from locally abundant flowering plants from March to August 2018. We targeted six unique species in one of four geographic regions of the United States (Supplementary Table 1; Fig. 1). The 24 total species represent 16 families (five subclasses), and multiple taxonomic groups were represented in each of the four regions (5 – 6 families/region, 2 – 4 subclasses/region). All the plant species were showy, herbaceous, and pollinated by animals (bees, flies, butterflies/moths, and birds; Supplementary Table 1; Fig. 1), but varied widely in inflorescence size, flower size, flower symmetry, and flower longevity, as well as in reward type and reward accessibility to pollinators (Supplementary Table 1). Moreover, their pollen grains varied in two traits (size and texture) important for collectability by pollinators (Supplementary Table 1). We reduced this phenotypic variation to four orthogonal principal components (PCs), of which PC1 – 3 accounted for ~65% of the total variation in floral and pollen grain phenotypes (Fig. 2a). Notably, PC1 captured variation in inflorescence size, flower symmetry, and floral reward accessibility; PC2 represented variation in pollen grain size and texture (Fig. 2a); and PC3 reflected differences in flower size and longevity (Supplementary Table 2). While the PCs displayed significant phylogenetic signals (all Pagel's $\lambda > 0.99$, $P = 0.003$ for PCs 1 – 2 and $P = 0.03$ for PC3), the plant species were well-distributed across floral and pollen grain trait space (PC1 vs PC2, Fig. 2a). Analyses of variance (ANOVAs) were used to determine whether PCs 1 – 3 varied between the regions. We found that PC1 significantly varied between the regions (df = 3, F = 3.23, $P = 0.04$), though a post-hoc Tukey's test revealed only modest differences between the California Grassland and California Coast ($P = 0.08$) regions and the Central Appalachian and California Grasslands regions ($P = 0.08$). There were no meaningful differences in PC1 between any other pairs of regions (all $P > 0.36$). PCs 2 and 3 did not vary significantly between the regions (df = 3, F = 1.59, $P = 0.22$ and df = 3, F = 1.93, $P = 0.16$, respectively).

The average land use patterns surrounding the sampled species varied among regions (Fig. 1). The California Coastal (CC) sites were characterized by low levels of human modification, mostly impervious surfaces (mean = 10%, SE = 3%) and high levels of natural vegetation (mean = 88%, SE = 3%), whereas the California Grassland (CG) sites were remote and on average experienced very little human-modified habitat (2%, SE = 1%), relative to natural vegetation (98%, SE = 1%). Within the Eastern Temperate Forest biome, the Central Appalachian (CA) sites were in preserved natural habitat in the Blue Ridge Mountains in North Carolina and Georgia with modest amounts of human-modified habitat (mean = 11%, SE = 5%), while the Eastern Deciduous Agro-forest Interface (EDAFI) sites in Pennsylvania were more strongly modified, with a high proportion of agricultural use (mean = 37%, SE = 4%). The variance in land use in each region was quite small, ranging from 0 to 0.025.

**Known viral taxa associated with pollen.** To identify pollen-associated viruses, total RNA extracted from the pollen of the 24 focal plant species was subjected to next-generation (i.e., high-throughput) sequencing. The resulting non-plant reads were directly aligned to viral nucleotide and protein sequence databases; the non-plant reads were also assembled into contigs, and

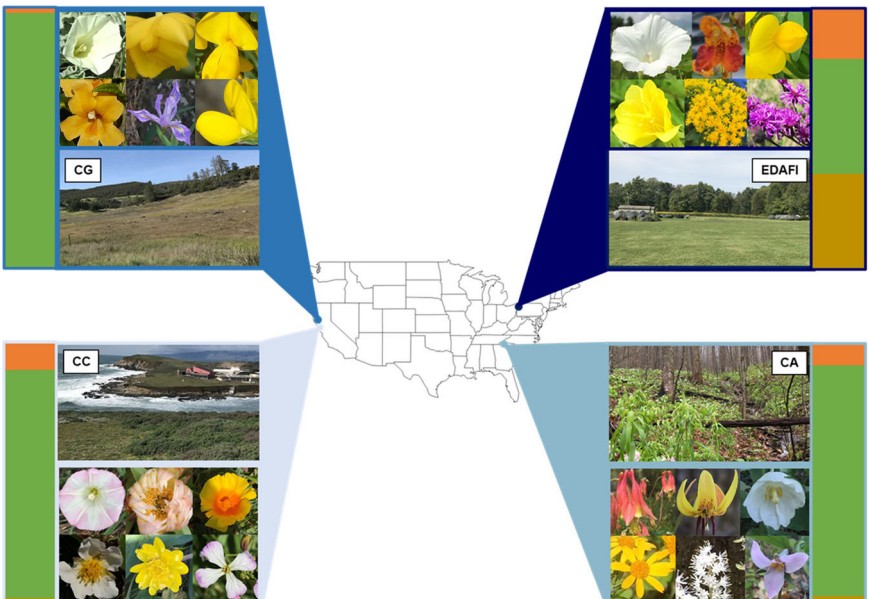

**Fig. 1 The four sampling regions and the 24 plant species studied (top left to bottom right).** The California Grasslands (CG): *Calystegia collina, Calochortus amabilis, Cytisus scoparius, Diplacus aurantiacus, Iris macrosiphon, Thermopsis macrophylla*; The Eastern Deciduous Agroforest Interface (EDAFI): *Convolvulus arvensis, Impatiens capensis, Lotus corniculatus, Oenothera biennis, Solidago sp., Vernonia gigantea*; The California Coast (CC): *Calystegia macrostegia, Carpobrotus edulis, Eschscholzia californica, Fragaria chiloensis, Ranunculus californica, Raphanus sativus*; The Central Appalachian forest (CA): *Aquilegia canadensis, Erythronium americanum, Podophyllum peltatum, Packera aurea, Tiarella cordifolia, Trillium grandiflorum*. Photos belong to the authors, or are from Creative Commons Images (*I. macrosiphon*: John Rusk, *C. scoparius*: John Haslam, *M. aurantiacus*: flickr profile "docentjoyce", CC BY 2.0 license; *C. amabilis*: David A. Hoffman, CC BY-NC-ND 2.0 license; *T. macrophylla*: Don Loarie, CC BY-NC-SA 2.0 license). Average land use percent cover for each region within a 3-km radius around the collection sites is indicated in the bar charts, where agriculture is represented in gold, urbanization (impervious surface) in orange, and natural vegetation (grassland and forest) in green. Source data are provided as a source data file.

open reading frames were aligned to protein sequence databases (Supplementary Table 3). We classified our sequences as known or novel viruses following the viral family-specific percent identity species demarcation criteria of the International Committee on Taxonomy of Viruses (ICTV)[13]. Viruses that were different from a known virus, but did not reach ICTV family-specific percent identity species demarcation thresholds, are identified hereafter as novel variants of known viruses. Novel viruses and novel variants were together analyzed further, see below. Viral sequences were detected in 23 of the 24 species-level pollen samples (Supplementary Table 3). No sequences related to viruses were found in association with *Erythronium americanum* (yellow trout lily) pollen.

We detected 22 known viruses, including 17 complete viral genomes, in pollen (Supplementary Table 4; Fig. 3). All but four of these (*Deformed wing virus, Alternaria arborescens mitovirus 1, Fusarium globosum mitovirus 1*, and *Hubei narna-like virus 25*) are classified as plant viruses, and only eight (*Apple mosaic, Prunus necrotic ringspot, Strawberry necrotic shock, Tobacco streak, Cherry rasp leaf, Tobacco ringspot, Tomato ringspot*, and *Deformed wing viruses*) have been previously described as being pollen-associated. Thus, our study added an additional 14 known viral species to the pollen virome. These include members of the Narnaviridae, Tombusviridae, and Tymoviridae, three viral families with no or little previously documented association with pollen, although some tombusviruses and tymoviruses are weakly seed-transmitted and thus could be pollen-transmitted as well (Supplementary Table 4). One of the non-plant viruses detected, *Deformed wing virus*, is a known bee pathogen that is transmitted to susceptible colonies via infected pollen[47]. While it is usually found on the outside of pollen grains, it may be tightly bound to their outer layer as well[47]. Two of the other non-plant viruses detected, *Alternaria arborescens mitovirus 1* and *Fusarium globosum mitovirus 1*, infect fungi[48].

Pollen from 11 of the 24 plants contained at least one known virus, and known viruses were detected in pollen from all regions except for the California Grasslands (Supplementary Table 4; Fig. 3). Of the 22 known viruses that we identified, 10 were found in association with the pollen of more than one plant species, and *Brome mosaic virus* was found in both sampling regions in the Eastern Temperate Forest biome (CA and EDAFI).

**Novel pollen-associated viruses**. We identified six coding-complete novel RNA viral genomes and three coding-complete novel variants of known viruses in association with pollen (Supplementary Table 5). They were found in the pollen of six plant species and from three of the geographic regions, although four were found in plant species from the Eastern Deciduous Agro-forest Interface (Supplementary Table 5; Fig. 3). They represent five viral families and include one novel species belonging to Amalgaviridae, two novel species belonging to Partitiviridae, three novel species and one novel variant belonging to Narnaviridae, and one novel variant each belonging to Bromoviridae and Secoviridae. In each case, the genome architecture matched key characteristics of the identified viral family (Supplementary Fig. 1). Furthermore, phylogenetic analyses placed all these novel genomes into known clades within these viral families (Fig. 4).

**Genetic signature analysis reveals novel partial pollen-associated viruses**. Most viruses in the Earth's virome are unknown, so it was not surprising that many virus-related sequences detected by our pipeline were novel. Based on the same criteria for coding-complete RNA viral genomes, we identified 203 novel partial genomes and variants (Supplementary Table 6). To confirm that these sequences were viral, we searched for and

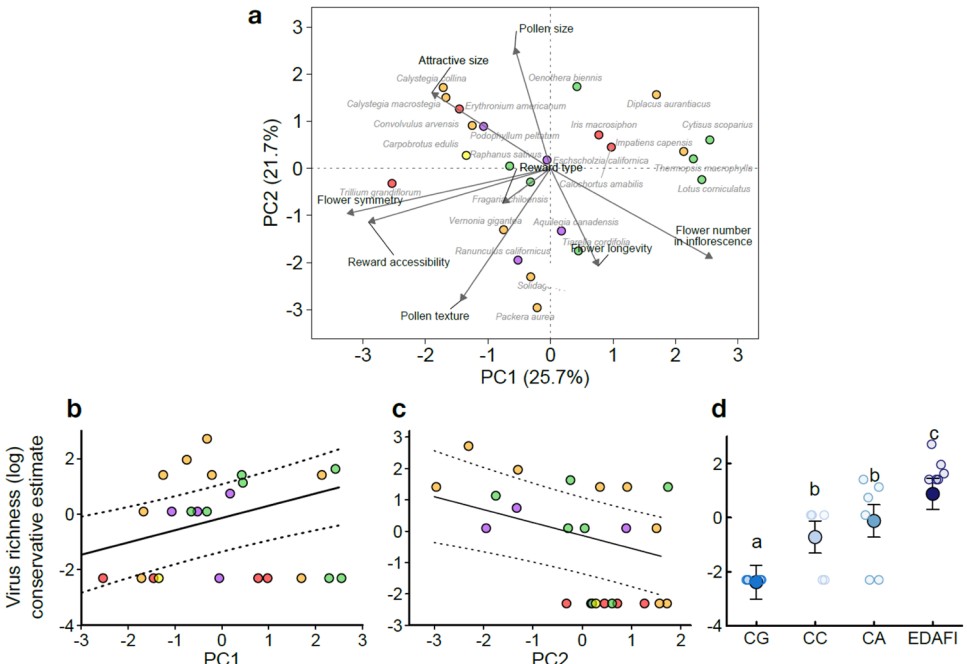

**Fig. 2 Surveyed plant species varied widely in floral and pollen grain traits, and these, along with geographic region, predicted the conservative estimate of virus richness in pollen. a** Plot of the first two principal components and associated floral loadings (black arrows and font) across plant species (grey font and colored dots). Direction of the arrows reflect the association between the higher binary value (1) of each trait category with PC1 or PC2. Plant species are shown as individual points, and colors represent the five plant subclasses: orange (*Asteridae*), yellow (*Caryophillidae*), purple (*Magnoliidae*), green (*Rosidae*), red (*Liliidae*). **b** Floral PC1, for which higher values reflect multiple-flowered inflorescences, bilateral floral symmetry, and restricted access to floral rewards, positively predicted the log-transformed conservative estimate of virus richness (two-sided chi-squared test, $P < 0.001$). **c** Floral PC2, for which lower values reflect spiky and smaller pollen grains, negatively predicted the log-transformed conservative estimate of viral richness (two-sided chi-squared test, $P = 0.001$). For **b** and **c**, the dotted lines represent 95% confidence intervals. **d** The log-transformed conservative estimate of viral richness in each region (two-sided chi-squared test, $P < 0.001$). CG = California Grasslands, CC = California Coast, CA = Central Appalachia, EDAFI = Eastern Deciduous Agro-forest Interface. $n = 24$ biologically independent pollen samples. Error bars represent ±1 standard error. P-values were adjusted using the Tukey method for comparing a family of four estimates. **a**, **b**, and **c** refer to statistically significant differences in the conservative estimate of virus richness between regions. Source data are provided as a source data file.

located key viral protein sequences (i.e., conserved domains) associated with RNA-dependent RNA polymerases (RdRps), coat proteins, genome- and membrane-linked proteins, movement proteins, proteases, Caulimoviridae viroplasmins (i.e., transactivator proteins), RNA silencing suppressors, Caulimoviridae RNases, aphid transmission factors, read through proteins, glycosyltransferases, helicases, methyltransferases, replicases, and reverse transcriptases.

The novel partial genomes and variants belong to 20 described viral families. Members of nine of the 20 described viral families have not been previously reported to be associated with pollen. We could not classify one partial novel viral genome beyond the order level (Ranunculus californicus mononegavirales 1) due to its lack of similarity to known viral families in NCBI databases, and 32 others did not belong to any known viral family or genus. Novel partial viral genomes and variants were found in all survey regions and in association with pollen of 22 of the 24 plant species (Supplementary Table 6). Like the novel complete-coding viral genomes and variants, many of the novel partial viruses were identified in association with pollen from plants in the Eastern Deciduous Agro-forest Interface.

**Pollen-associated viruses are heterogeneously distributed across viral families and plant subclasses.** The known viruses and coding-complete novel viral genomes and novel variants represent nine described viral families; however, over half of them belong to three viral families: the Bromoviridae, Partitiviridae, and Secoviridae (Supplementary Tables 4–5; Fig. 3). A permutation test revealed that

this distribution of viruses is significantly different from random chance, providing confirmatory evidence that these viral families have members with characteristics that allow for the exploitation of the pollen niche (observed = 39, 95% CI = 27 – 38, P < 0.05). When considering the more comprehensive (relaxed) estimate of virus richness that also included certain novel partial viral genomes (i.e., RdRps) in addition to the conservative estimate of virus richness, the pattern remained the same, though it was not significant (observed = 79, 95% CI = 59 – 80, $P > 0.05$).

Neither the conservative nor relaxed virus richness estimates were significantly influenced by plant evolutionary history (Pagel's λ = 0.35, 0.42, respectively; $P = 0.34, 0.31$, respectively). However, the known viruses, novel coding-complete viral genomes, and novel variants were not evenly distributed across the five plant subclasses (Fig. 3). Most were found in pollen from the *Asteridae*, to which *Packera aurea*, the *Solidago* sp., and *Vernonia gigantea* belong. A permutation test indicated that this viral distribution is significantly different from random chance (conservative: observed = 35, 95% CI = 11 – 24, P < 0.05; relaxed: observed = 87, 95% CI = 37 – 58, P < 0.05).

**Ecological correlates of virus richness.** Several floral and pollen grain traits were significant predictors of virus richness. Pollen from plant species with multiple-flowered inflorescences, bilateral floral symmetry, and restricted access to floral rewards had higher conservative virus richness estimates than plant species with single, radially symmetric flowers with easily accessible rewards (PC1: $\chi^2 = 13.77$, df = 1, $P < 0.001$, Fig. 2a, b), although this

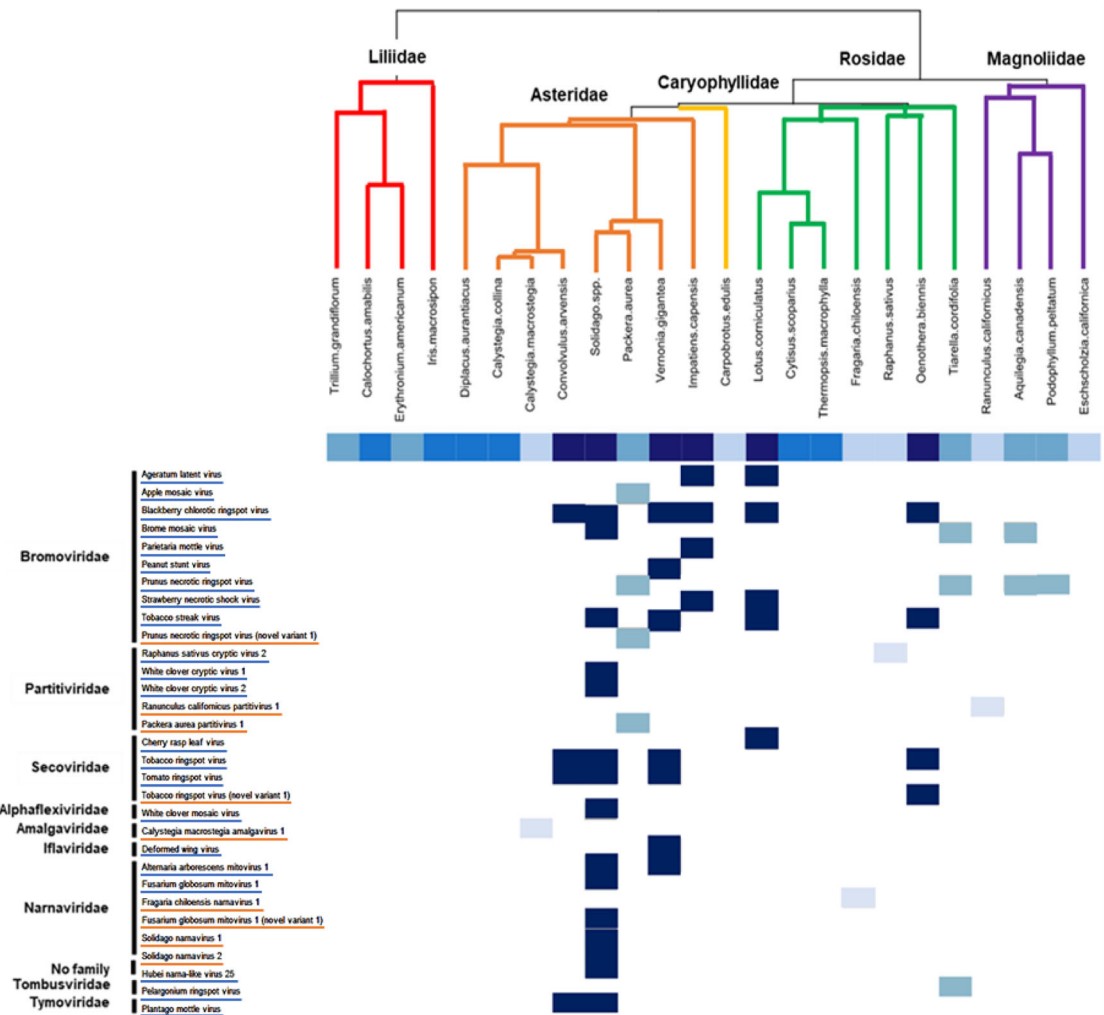

**Fig. 3 Pollen-associated known viruses, novel coding-complete viral genomes, and novel variants of known viruses grouped by family (right), plant species (top) and geographic region (central boxes).** Known viruses (blue underline), are referenced by their species name. Novel coding-complete viral genomes and variants (orange underline) are referenced by the plant species in which they were discovered and the virus family to which they belong. Plant subclasses are indicated by color on the dendrogram. Presence of a virus in a plant species is indicated by a box with the color indicating the geographic region of collection (navy blue = EDAFI, medium blue = CA, light blue = CC). Source data are provided as a source data file.

pattern did not persist when considering the relaxed estimates of virus richness ($\chi^2 = 1.46$, df $= 1$, $P = 0.23$). Plant species with smoother or larger pollen grains harbored significantly lower virus richness estimates than plant species with spiky or smaller pollen grains (PC2: conservative: $\chi^2 = 11.33$, df $= 1$, $P = 0.001$, Fig. 2a, c; relaxed: $\chi^2 = 3.73$, df $= 1$, $P = 0.053$, Supplementary Fig. 2a). Plant species with larger and longer-lived flowers (PC3), however, did not have significantly higher pollen virome richness estimates than plant species with smaller, shorter-lived flowers (conservative: $\chi^2 = 2.05$, df $= 1$, $P = 0.15$; relaxed: $\chi^2 = 0.002$, df $= 1$, $P = 0.96$).

After accounting for the influence from floral and pollen grain traits and plant evolutionary history, virus richness varied significantly among the four geographic regions (conservative: $\chi^2 = 55.19$, df $= 3$, $P < 0.001$, Fig. 2d; relaxed: $\chi^2 = 17.66$, df $= 3$, $P < 0.001$, Supplementary Figure 2b). The region with the highest proportion of human-modified land use (Fig. 1)—the Eastern Deciduous Agro-forest Interface—had the highest virus richness (Fig. 2d and Supplementary Fig. 2b), especially compared to the natural vegetation-dominated California Grasslands region, where only 2% of land was human-modified (post-hoc LSmeans contrast, conservative: $t = -7.23$, df $= 14.85$, $P < 0.001$, Fig. 2d;

relaxed: $t = -3.08$, df $= 13.2$, $P = 0.007$, Supplementary Fig. 2b). Overall, virus richness was positively correlated with increased human-modified land use and decreased natural vegetation (conservative: Spearman's $\rho = 0.80$, $P = 0.20$, $N = 4$; relaxed: Spearman's $\rho = 1.00$, $P < 0.001$, $N = 4$).

## Discussion

We used a metagenomic pipeline to define the pollen virome in a taxonomically and geographically diverse collection of visually asymptomatic, wild plant species. Fourteen of the known viral species that we identified were not previously recognized to be associated with pollen and thus this study significantly expands knowledge of the pollen virome. Furthermore, the coding-complete genomes of six novel viral species and three novel variants, as well as the partial genomes of many novel viral taxa and variants, were identified as pollen-associated. Many viruses previously reported to be pollen-associated belong to the Bromoviridae, Secoviridae, and Partitiviridae viral families. Our work confirms this pattern, but we also found viruses belonging to several other viral families not previously known to be pollen-associated. We found that plant species with traits that promote increased plant-pollinator vector interactions and those

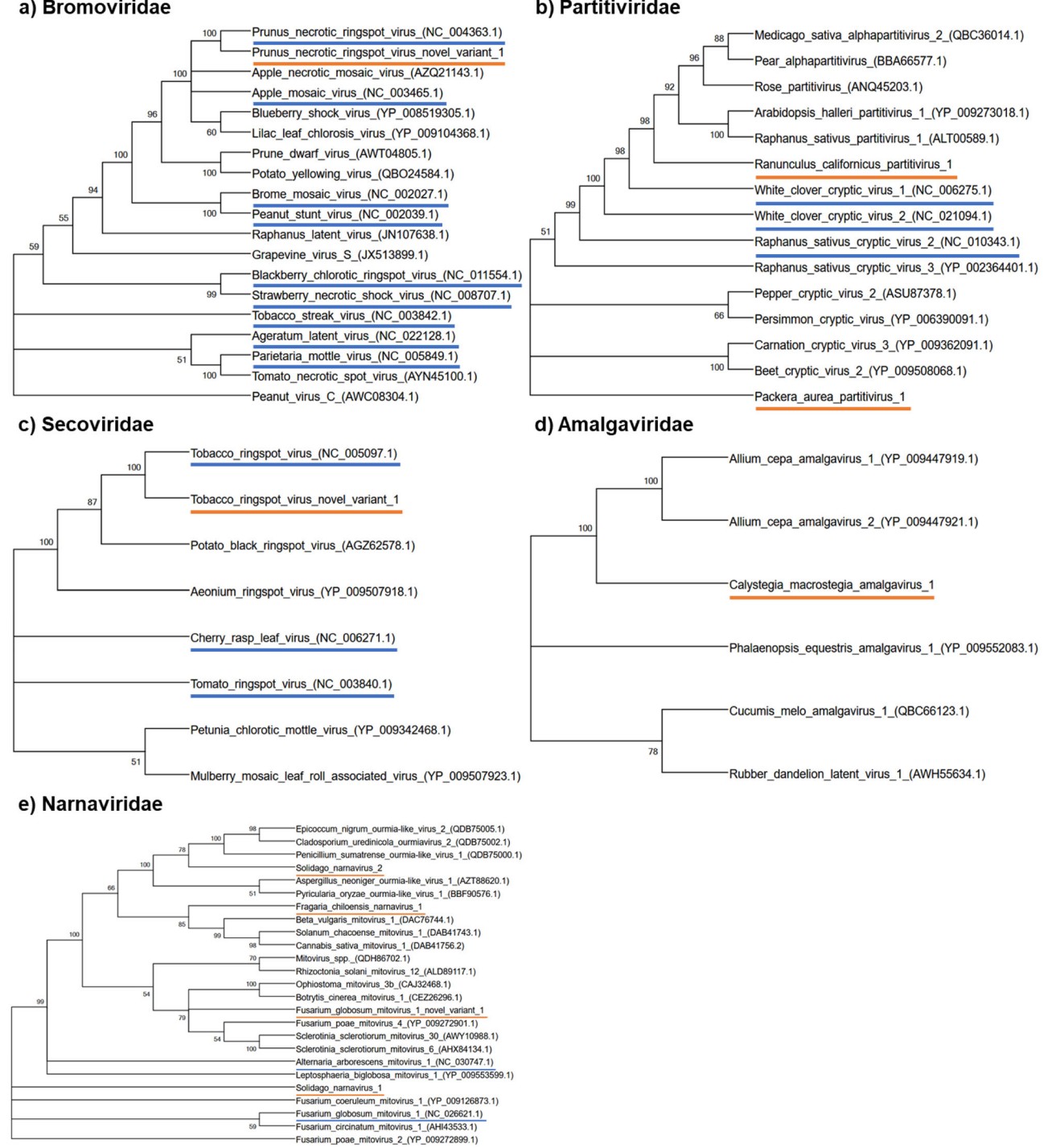

**Fig. 4 Maximum-likelihood bootstrap consensus phylogenies of viruses based on amino acid sequences of the RdRp region are presented by family.**
**a–e** Known viruses (blue underline) and novel viral genomes and variants of known viruses (orange underline) found in association with pollen are presented along with taxa that represent BLASTP hits to the RdRp regions of the novel viral genomes. Bootstrap support values from 500 replicates are shown at the nodes. Source data are provided as a source data file.

surrounded by more land impacted by humans and with less natural vegetation had the highest virus richness. These features, along with plant subclass, had significant predictive power in describing the distribution of virus richness, providing a first glimpse at the potential ecological drivers of this unique viral niche, and setting the stage for future, finer-scale dissections of the mechanisms behind these species-level patterns.

The richness of the virome was significantly influenced by the region from which pollen was collected. While controlling for the

phylogenetic distribution of plant species, we found that the greatest number of viruses overall was found in pollen from species growing in the Eastern Deciduous Agro-forest Interface, a region where land use patterns tip heavily toward human modification at the expense of undisturbed natural vegetation (Fig. 1). Interestingly, we also found more known viruses in association with pollen from the Eastern Deciduous Agro-forest Interface (Supplementary Table 4) and many novel viruses in association with pollen from the California Coast and Central Appalachian

regions (Supplementary Tables 5–6). This pattern may reflect the facts that many known plant viruses were identified from agricultural crops rather than wild species and that the plants in the Eastern Deciduous Agro-forest Interface are surrounded by more agricultural land use than the plants in the California Coast and Central Appalachian regions. Although many plant viruses, including pollen-associated ones, were originally described in plants of agricultural importance, many can infect wild plants[28,49], and our results provide evidence of the wider host ranges of these viruses. In fact, if the viral diversity–land use patterns seen here are due to human disturbance and the potential for viral spillover rather than reflecting viral host range, then we would predict significant variation in pollen-associated virus incidence or diversity within and among populations, dependent upon proximity to agriculture or other human-disturbed habitats. Deeper sampling within a focal species and across a range of habitat types is needed to test this hypothesis. Furthermore, this alone would not be sufficient to demonstrate that pollinators are the important vectors transmitting viruses from cultivated to wild plants and vice versa[28,31] because several of the viral taxa identified could also be transmitted by herbivorous insects (e.g., *Brome mosaic virus*[50]). Thus, more detailed sampling of pollinator-collected pollen[5,20], paired with plant-level and pollen grain-level analyses, as well as herbivore exclusion, is needed to substantiate the role of pollinators as key vectors across land use gradients.

Our study uncovered a previously unrecognized relationship between pollen virome richness and plant traits that attract pollinators and influence their interactions with pollen, thus by extension highlighting the potential importance of pollinators as vectors of pollen-associated viruses. Multiple flowered-inflorescences increase the likelihood that plants will interact with pollinators in general and increase the likelihood that a plant will interact with diverse flower visitors[e.g.,51–53]; here, we found that they also positively predicted the taxonomic richness of the pollen virome. We found that plant species with bilaterally symmetric flowers had richer viromes, suggesting that restricted and directed pollinator access may lead to more contact with the stigma and increased transfer of pollen and pollen-associated viruses. We also found that having spiky or smaller pollen grains significantly and positively predicted pollen virus richness. Although once thought to impede pollen grain collectability[e.g.,54], these traits may actually benefit virus transmission to a plant because a spiky exine does not necessarily prevent pollen from being collected by bumble bees[55–57] and may even help pollen cling to pollinators, thus enhancing transfer. Viruses and other bioparticles may also become trapped on spiky exines, further aiding in virus transmission. Likewise, since smaller pollen grains are potentially easier for pollinators to handle and pack into pollen loads[e.g.,55,56], pollinators as vectors of pollen-associated viruses may preferentially visit plant species that produce smaller grains. Our survey has opened the door for future investigations of the causal aspects of these associations, including those where the location of individual plants or their floral traits are manipulated and the response in the pollen virome is recorded. Our survey also begs the question of what other pollen traits might be associated with virus richness? For instance, pollen traits that affect viral infection, persistence, or transmission to pollen within a host would be good targets for future investigations. Finally, our study opens the possibility that the evolution of floral and pollen traits themselves are shaped by viral pathogens, not just pollination, as observed for other plant antagonists[58]. If this is the case, we might expect plant species with similar traits to share similarly diverse pollen viromes.

Since we could not distinguish the location of the viruses in the pollen grains sampled, it is possible that the presence of them is due to casual contact with other hosts that contacted the plants. For instance, the pollen sample that harbored *Deformed wing virus*, a bee-infecting pathogen known to be transmitted to susceptible colonies via pollen, may be an example of transient contact between pollen in anthers and infected bees, although this virus can also be tightly bound to the outer layer of pollen grains[47]. We detected thousands of reads that aligned to some viruses, and many assembled contigs were present in high abundance (Supplementary Tables 4–6), but detected no common environmental contaminants. Together, these observations suggest that some viruses were not merely molecular hitchhikers or contaminants, but instead were actively infecting the surveyed plants, even though the plants did not exhibit noticeable signs of disease.

Our study demonstrates, not only that pollen is a unique viral niche, but also that it can host a diverse set of viral taxa. Viruses from the Bromoviridae, Partitiviridae, and Secoviridae families were common in pollen, perhaps indicating that their characteristics (e.g., vertical and horizontal infection routes, acute lifestyles, and movement and coat proteins[59]) may allow them to exploit the pollen niche. The identification of several plant traits that increase the association between plants and their pollinators, who are important pollen-associated virus vectors, as well as the land use patterns correlated with virus richness, expand our knowledge of viral host ranges and recognize for the first time the diversity of viruses that could be pollinator-transmitted. The prevalence of pollen-associated viruses across the plant families and subclasses we sampled suggests that we are only beginning to understand pollen as a viral niche and that it is ripe for continued research on finer-scale patterns of infection (i.e., among populations, within populations, and within individuals), as well as on function. If found to be prevalent, then pollen-associated viruses may threaten plant biodiversity and food security more widely than previously recognized.

## Methods

**Pollen collection and RNA extraction.** Pollen is a microscopic and notoriously resistant plant product. Thus, methods to collect a sufficient and roughly equivalent volume of pollen per species, and to ensure RNA was collected from viruses both internal and external to pollen grains, were developed specifically for this work. At each of the four regions, we identified visually asymptomatic plants species that were in full flower and in high enough abundance to achieve our pollen sample minimum. Many of the pollen samples were collected from public roadsides. However, some from the California Grasslands were collected from the University of California's McLaughlin Natural Reserve, and some from the Eastern Deciduous Agro-forest Interface were collected from the University of Pittsburgh's Pymatuning Laboratory of Ecology. We had permission to sample in both places. In addition, we obtained permission from the USDA Forest Service to sample in the Till Ridge Cove area of the Chattahoochee-Oconee National Forest for sampling in Central Appalachia. None of the sampled plants displayed classic viral symptoms (e.g., leaf yellowing, vein clearing, leaf distortions, growth abnormalities). To achieve the broadest representation of plant species, we selected species in different families, where feasible. Also when possible, we focused primarily on perennial species to avoid any effects of life history variation. From these, we collected 30 to 50 mg of pollen from newly dehiscing anthers (3–967 fresh hermaphroditic flowers from 1–27 plants per species; Supplementary Table 3) in situ using a sterile sonic dismembrator (Fisherbrand Model 50, Fisher Scientific, Waltham, MA, USA) with a frequency of 20 Hz. We removed non-pollen tissues (e.g., anther debris) with sterile forceps. In addition to removing non-pollen debris that was visible to the naked eye in the field at the time of pollen sample collection, we conducted microscopic and gene expression analyses to confirm the purity of the pollen samples in the lab (Supplementary Methods). Visibly pure pollen from a single species was transferred to a 2-mL collection tube with Lysing Matrix D (MP Biomedicals, Irvine, CA, USA) and kept on dry ice until transported to and stored at −80°C at the University of Pittsburgh (Pittsburgh, PA, USA).

Before extracting the total RNA, we freeze-dried the pollen samples (FreeZone 4.5 Liter Benchtop Freeze Dry System, Labconco Corporation, Kansas City, MO, USA) and lysed with a TissueLyser II (Qiagen, Inc., Germantown, MD, USA) at 30 Hz with varying times for different plant species (Supplementary Table 3). We confirmed via microscopy that this protocol resulted in the breakage of ≥50% of the pollen grains in a sample. The total RNA, including dsRNA, was extracted using the Quick-RNA Plant Miniprep Extraction Kit (Zymo Research Corporation, Irvine, CA, USA), following the full manufacturer's protocol, including the optional steps of in-column DNA digestion and inhibitor removal.

**RNA sequencing.** We assessed the quantity and quality of the total RNA extracted from each pollen sample with a Qubit 2.0 fluorometer (Invitrogen, ThermoFisher

Scientific, Waltham, MA, USA) and with TapeStation analyses performed by the Genomics Research Core (GRC) at the University of Pittsburgh. Only samples with an RNA integrity value of ≥1.9 were used (Supplementary Table 3). Stranded RNA libraries were prepared by the GRC using the TruSeq Total RNA Library Kit (Illumina, Inc., San Diego, CA, USA), and ribosomal depletion was performed using a RiboZero Plant Leaf Kit (Illumina, Inc., San Diego, CA, USA). At the GRC, we pooled depleted RNA libraries from six species on a single lane of an Illumina NextSeq500 platform.

**Pre-virus detection steps**. A sequencing depth of 117–260 million 75 bp paired-end reads was achieved per sample (Supplementary Table 3). Sequences were demultiplexed and trimmed of adapter sequences. We used the Pickaxe pipeline[42,60,61] to detect known and novel pollen-associated viruses. First, Pickaxe removes poor-quality raw reads[42,60,61] and aligns the quality-filtered reads using the Bowtie2 aligner with default parameters[62] to a subtraction library. Each customized subtraction library contained the host plant species genome or the most closely related plant genomes in the National Center for Biotechnology Information (NCBI) database, if the host plant genome was not available (Supplementary Table 7), as well as other possible contaminant genomes (e.g., the human genome)[42,60,61]. The subtraction libraries with 1–8 closely related plant genomes, a bioinformatically tractable amount, were used to remove plant sequences, which allows for a conservative estimate of the viruses associated with pollen to be made. The size of the subtraction libraries did not influence the number of identified viruses, as there was no correlation between library size and either estimate of virus richness (conservative: $r = 0.08$, $P = 0.75$; relaxed: $r = 0.06$, $P = 0.77$). After subtraction, only non-plant reads remained and were used for viral detection.

**Known RNA virus detection, identity confirmation**. With Pickaxe, we used the Bowtie2 aligner with default parameters[62] (v2.3.4.2-3) to align viral non-plant reads to Viral RefSeq[42,60,61] (hereafter, VRS; Index of /refseq/release/viral (nih.gov)). Each known virus reflects the top hit of an alignment to VRS[42,60,61]. Following Cantalupo et al.[42], we considered a known virus to be present if the viral reads covered at least 20% of the top hit and aligned to it at least ten times. For viruses with segmented genomes, at least one segment was required to meet these criteria.

**Contig annotation and extension; novel RNA viral genome detection, identity confirmation**. Viral reads were assembled into contigs using the CLC Assembly Cell (Qiagen Digital Insights, Redwood City, CA, USA), and Pickaxe was used to remove repetitive, short (<500 base pairs), and heavily masked sequences[42,60,61]. Contigs that passed these quality steps were annotated following Starrett et al.[61], except that the GenBank nucleotide database was also searched with Rapsearch2[63] (v2.22). We then we used BLASTN (NCBI) to search for overlapping regions that were at least 90% identical between contig ends to extend them, if possible.

Main criteria used to confirm the identification of novel RNA viruses (genomes and variants) were: (1) contig or extended contig length corresponded to a putative viral family[13]; (2) the dissimilarity of a contig or extended contig from the top BLAST or RAPSearch2 hit exceeded the percent identity threshold for its putative family, as per the ICTV species demarcation criteria[13]; (3) the open reading frame (ORF; https://www.ncbi.nlm.nih.gov/orffinder/, default parameters, v0.4.3) architecture of a contig or extended contig matched that (or part of that) of a putative viral family[13,64]; and (4) detection of at least one of the conserved viral domains (i.e., proteins; hereafter, CDs) in the ORFs of a contig or extended contig with a search of the Conserved Domain Database[65] (https://www.ncbi.nlm.nih.gov/Structure/cdd/wrpsb.cgi, default parameters, v3.16-17) corresponding to the CDs of a putative viral family[13]. We also considered contig relative abundance (i.e., the number of reads assembled into a contig divided by the contig length; representing the overall number of reads belonging to a novel viral genome) and how much of a contig participated in the alignment with the top BLAST or RAPSearch2 hit. Novel coding-complete genomes or variants of known viruses (i.e., those that met all or nearly all the above criteria) are reported at the family level. Genome organization and coverage depth across all these are shown in diagrams drawn to a unified length scale and depth plots, respectively (Supplementary Fig. 1). Depth plots were created using Bowtie2[62] to align the non-plant reads to contigs and Samtools[66] (v1.9) to determine coverage depth at each base. In addition to novel coding-complete viral genomes and variants of known viruses, we also report novel partial RNA viral genomes and variants. All novel viral genomes were named by the plant species in which they were discovered, the putative viral family, and a number, and novel viral variants were named after their known viral species name.

**Virus richness estimation and correlations**. For each pollen sample, we calculated the conservative virus richness, or the total number of known viruses, novel coding-complete viral genomes, and novel coding-complete variants. We also calculated the relaxed estimate of virus richness, which also included the novel partial RNA-dependent RNA polymerase (RdRp) CDs of both genomes and variants. Since we collected the same volume of pollen from all plant species, we determined whether sampling variation was related to virus recovery by correlating virus richness estimates with the number of individuals and flowers sampled.

We found that the conservative and relaxed estimates of virus richness were highly correlated ($r = 0.96$, $P < 0.001$), and both were correlated with the number of

flowers (both $r > 0.74$, $P < 0.001$), but not the number of individuals (conservative: $r = -0.24$, $P = 0.26$; relaxed: $r = -0.27$, $P = 0.20$) sampled. These patterns were unaffected by removing outliers (i.e., plant species where >100 flowers were sampled; correlations with flowers sampled: conservative $r = 0.49$, $P = 0.02$; relaxed: $r = 0.40$, $P = 0.05$; correlations with individuals sampled: conservative: $r = -0.18$, $P = 0.40$; relaxed: $r = -0.23$, $P = 0.28$). To be conservative, however, we controlled for both flowers and individuals sampled in all the analyses of virus richness by adding them as covariates to the phylogenetically corrected linear models (see "*Flower, pollen traits…*" below).

**Plant and viral phylogenies**. We constructed a phylogeny of the plant species based on the PhytoPhylo maximum likelihood megaphylogeny of vascular plants[67,68] with the R (v4.0.1) packages "ape" and "phytools"[69,70]. The positions of the two plant species that were not present in the megaphylogeny data set (*Calochortus amabilis* and *Calystegia collina*) were manually added to the tree according to genus-level phylogenetic relationships[71,72].

To assess the taxonomic membership of coding-complete novel viral genomes and variants and known viruses, we built maximum-likelihood family-level viral phylogenies by first aligning the amino acid sequences of the novel coding-complete viral genome RdRp CDs using the MUSCLE algorithm, with default parameters in MEGA X[73]. We then ran 500 bootstrap replicates of the Jones-Taylor-Thornton matrix-based model with default parameters. In doing so, we applied the Neighbor-Join and BioNJ algorithms in MEGA X to a model-generated matrix of pairwise distances between each sequence, and the topology with the best log-likelihood value is reflected in the phylogenies[73–75]. Following Galbraith et al.[76] to create a frame of reference in these phylogenies, we also included the top five unique BLASTP (NCBI) hits with the closest percentage identity to each RdRp sequence.

**Pollen-associated RNA virus distribution**. To assess the evolutionary dependence of virus richness among the plant species (the conservative and relaxed estimates separately), we tested for a phylogenetic signal using Pagel's λ[77] with the R package "phytools"[70], and concluded a phylogenetic signal was present if Pagel's λ was significantly above zero. We evaluated whether the conservative and relaxed estimates of virus richness were disproportionally distributed among the five plant subclasses by creating null models that assumed random distribution of the viruses among the plant species and shuffling virus presence ($N = 1000$) using the R package "vegan"[78]. To assess significance, we compared the observed virus richness of each plant subclass to its 95% null confidence intervals.

To assess whether the viruses included in the conservative and relaxed estimates of virus richness belonged disproportionately to the Bromoviridae, Partitiviridae, and Secoviridae viral families, we created null models that assumed a random virus distribution across all viral families and shuffled virus presence among them ($N = 1000$), as above. We compared the combined observed virus richness in the Bromoviridae, Partitiviridae, and Secoviridae viral families to the 95% null confidence intervals of the same group.

We visualized known virus and novel coding-complete viral genome and variant distribution across plant species, viral families, and geographic regions using the R packages "gplots," "Heatplus," and "RColorBrewer"[79–81].

**Flower, pollen traits, and land use as drivers of pollen-associated virus richness**. To assess whether floral traits explained variation in virus richness, we recorded traits important for pollinator vector attraction (inflorescence type, flower longevity, flower size [or equivalent floral unit of attraction], floral rewards) and floral reward accessibility (flower symmetry and accessibility based on floral morphology) as described in the literature (see Supplementary Table 1). In addition, we scored two traits important for pollen grain collectability: pollen grain texture and size (diameter of the longest dimension in μm), which were determined with the aid of a light microscope (magnification 10X or 40X; Leica DM500, Leica Microsystems, Buffalo Grove, IL, USA).

Prior to analysis, we coded the levels of categorical traits as 0 or 1 as follows: the number of flowers in the inflorescence (single vs. multiple [including cyme, raceme, panicle, and heads]), rewards (pollen only vs. pollen and nectar), flower symmetry (bilateral vs. radial), reward accessibility (restricted [by morphology or time] vs. accessible), and pollen grain texture (granulate [all non-spiky] vs. echinate [spiky]). All traits were standardized (i.e., mean = 0, standard deviation = 1).

We performed a principle component analysis (PCA) on all eight floral traits (three quantitative and five ordinal, binary) described above, which yielded three dominant floral trait principal components (PC1 – 3) using the "prcomp" function in R. The PCA results were validated using a factor analysis of mixed data (FAMD) for quantitative and qualitative variables, implemented in the package "FactoMineR"[82], which yielded results identical to those of the PCA. To assess whether floral traits reflected shared evolutionary histories among plant species, we tested for phylogenetic signals of PC1 – 3 using Pagel's λ in the R package "phytools"[70]. We then evaluated which floral traits influenced the conservative estimate of virus richness, while accounting for the influence of geographic region, with a phylogenetically corrected linear model using the R package "nlme"[83]. To improve normality, we added a small constant (0.1) to the conservative estimate of virus richness prior to natural logarithm transformation. The predictors of the model included the first three floral PCs and region, with the number of flowers and individual plants sampled as covariates to account for potential variation

in virus recovery. This nested linear model design, and the explicit inclusion of the phylogenetic relationships among the plant species, allowed us to treat the plant species as replicates in each region and to isolate the effects of the floral traits, while controlling for the evolutionary history of the plant species. Variance inflation factors implemented in the R package "car"[84] were used to confirm the absence of multicollinearity. We assessed the statistical significance of the predictors using type III sums of squares in "car" and estimated the least-squares means (LSmeans) using the package "emmeans"[85]. We repeated all statistical analyses with the relaxed estimate of virus richness, and all statistical analyses were performed in R (v4.0.1).

To characterize land use for each of the geographic regions, we circumscribed buffer zones with a 0.5 km- or a 3 km-radius around the spatial location of each plant species in each region using ArcGIS Desktop (v10.7.1; Environmental Systems Research Institute, Inc.). The larger radius reflects the average foraging distance of honey bees[86], which are common pollinators throughout much of the United States and in our landscapes. The foraging distances of other pollinators are also encompassed by the smaller buffer zone. To most accurately quantify the land use in each region, the land use percent cover within each circular buffer zone was calculated and averaged for three categories—agriculture, urban (impervious surface: buildings, sidewalks, roads, other hard surfaces), and natural vegetation (grassland and forest)—extracted from the National Geospatial Data Asset (NGDA) Land Use Land Cover dataset (v2014; NLCD Land Cover Change Index (CONUS) | Multi-Resolution Land Characteristics (MRLC) Consortium), which contains land use states for the years 2004 – 2011 at $0.30 \times 0.30$ degree spatial resolution. The estimates of land use within the buffer zones were highly correlated ($r = 1.00$, $P < 0.001$), so we present results only for the 3 km-radius buffer zone throughout the manuscript.

**Reporting summary**. Further information on research design is available in the Nature Research Reporting Summary linked to this article.

## Data availability
The raw reads were deposited in GenBank under Bioproject number PRJNA589022 and will be publicly available upon publication. The Pickaxe output from viral read alignments to VRS and the Pickaxe output from viral contig alignments to the GenBank nucleotide and protein databases are included as Supplementary Datasets 1 and 2, respectively. All contig sequences are also included in Supplementary Dataset 2. In addition, supplementary information and source data are provided with this paper.

## Code availability
Any code for the Pickaxe pipeline is available upon request or is accessible in Github (https://github.com/pcantalupo/pickaxe)[87]. Additionally, any R code generated for this project is available upon request.

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

## Acknowledgements

This research was supported by funding from the Charles E. Kaufman Foundation of the Pittsburgh Foundation (grant KA2017-91786 to TLA and JMP), the Leasure K. Darbaker Prize in Botany (University of Pittsburgh Pymatuning Laboratory of Ecology to AMF), and the Andrew Mellon Predoctoral Fellowship (University of Pittsburgh Kenneth P. Dietrich School of Arts and Sciences to AMF). The research was also supported in part by the University of Pittsburgh Center for Research Computing through the resources provided. We thank the staffs at the University of California McLaughlin Natural Reserve, the University of Pittsburgh Pymatuning Laboratory of Ecology, and the University of Pittsburgh Genomics Research Core. We also thank Dr. Martin Turcotte for the use of his freeze dryer and Abigail Jarrett and Nevin Cullen for field assistance.

## Author contributions

T.L.A. and J.M.P. designed the experiment; A.M.F., T.L.A., J.D.S., A.M.S., P.G.C. and M.T.S.R. collected the data; A.M.F., N.W., A.M.S., M.T.S.R. and P.G.C. analyzed the data; A.M.F., J.D.S., A.M.S., N.W., M.T.S.R. and P.G.C. produced the figures; A.M.F., N.W., J.M.P. and T.L.A. wrote the first draft, all authors contributed editorially to the manuscript, and author order was determined on the basis of these duties.

## Competing interests

The authors declare no competing interests.
