## [Peer Review File · Nature Communications]

Reviewer comments, initial review -

Reviewer #1 (Remarks to the Author):

The manuscript entitled: « Proximity to human disturbance and flower traits shape the pollen virome» is a very well-written manuscript that further explores virus ecology. Specifically, this manuscript aimed at deciphering the diversity of pollen virome and further tried to disentangle landscape and floral factors that may have impacted the diversity and distribution of pollen viruses.

The major claims of the ms. are that the pollen-associated virus diversity is much higher than previously thought and that both landscape (level of fragmentation by agriculture and urbanization) and plant traits (e.g. multiple flowered-inflorescences) are potentially playing a key-role in the taxonomic richness of the pollen virome and its geographic distribution.

While the conclusion of the study is original and interesting to others in the community, I have two major concerns about this manuscript:

1- Pollen purity. A specific protocol was developed for this study. Pollen was collected in situ using a sterile sonic dismembrator and non-pollen tissues (e.g., anther debris) were removed with sterile forceps. “Pure pollen” was then transferred to a collection tube with Lysing Matrix D that was kept on dry ice until being stored at -80°C at the University of Pittsburgh. I have concern about the purity of the pollen that was collected using this protocol, because only visible plant tissues were removed. The purity of pollen is crucial in this study while the authors are concluding that all viral contigs that were assembled were “pollen-associated”. This statement may actually be wrong because some of these viruses were potentially associated with plant debris (anther, leaf, etc.) that were probably contaminating the pollen fraction. To address this point, I would have suggested to add several controls. I’m not an expert of pollen purification but isolation and purification of highly purified, hydrated pollen grains can be achieved. Then, highly purifying a subset of the 145 plants that were analyzed in this study could have been carried out, for instance through flow cytometric sorting (e.g. Becker et al. (2003) *Plant Physiol.* 133, 713–725). In addition, I would have suggested to add plant controls, i.e. analyzing in parallel the plant virome of the plants from which the pollen was collected (again from a subset of the 145 plants that were analyzed in this study). This would have allowed distinguishing between viruses that were present in both pollen and plant viromes (and see whether they were phylogenetically related) and viruses that were only present in the pollen virome. The authors were aware about this problem of purity of the pollen while they have written in the discussion section “it is possible that the presence of the pollen-associated viruses is due to casual contact with other hosts that contacted the plants”.

2- Influence of the region from which pollen was collected. While I think that this is

convincing that there exists a previously unrecognized relationship between virus richness and plant traits, I have serious doubts as to whether the role of landscape (level of fragmentation by agriculture and urbanization) is relevant. Only one site (EDAFI) out of four was heavily fragmented by agriculture and urbanization. The EDAFI site represented only 18/145 samples (12.4%) that were processed. This site was compared to remote “wild” sites, which may have biased the comparison. I would have suggested to compare several neighboring “natural” and “disturbed” settings from the EDAFI region (and potentially repeat this design from other regions) to address this question of the role played by human-mediated disturbance on the diversity and distribution of the pollen virome.

Reviewer #2 (Remarks to the Author):

The manuscript “Proximity to human disturbance and flower traits shape the pollen virome” surveys plant species across landscape types to examine if plant species or environments are associated with the diversity viruses found in pollen. This topic is important and timely, and likely of very broad interest to pollination biologists, plant virologists and within ecology and evolution. Moreover, a broad survey of many plant species for viral presence in pollen has not been conducted previously so this study adds much to our understanding of this type of interaction.

The data presented here are novel, very interesting, and are surely to spur a lot of additional research on the topic. Congrats to the authors on this nice work. My main concern with the paper is that the conclusion regarding the link between disturbance/agriculture and virome structure is not adequately tested or supported. A few other methodological and statistical details are missing, as well as justification for the inclusion of the specific pollen traits included. As a result, I suggest a reframing of the paper to focus on its strength and what seems like its real contribution: describing the virome of a number of plant species which have not been previously examined. This contribution raises many ecological and evolutionary questions and will certainly spur lots of additional research on this topic. I detail the basis of my critiques below:

First, the data cannot address the main conclusion of the paper—that proximity to human disturbance and flower traits shape the pollen virome. This main message is also emphasized in the abstract (L 34-40) as a main finding of the paper. However, this conclusion is conceptually vague and not adequately tested. First, due to the sampling design used, the connection between human disturbance and viral composition or diversity cannot be directly made. This is because only a single sample from each plant species was included. Each of the plant species were sampled from a single bioregion, and as a result, plant species (and bioregion) is entirely confounded with human disturbance and so these potential predictors cannot be disentangled from each other. I could not find a description of the model related

viral richness to human disturbance (Lines 214-215), so I do not know if plant species was accounted for in this model.

The second issue with this conclusion is that it is conceptually vague: the introduction states that wild plants growing in disturbed areas may be more prone to viral spillover from crops. However, the paper does not specifically address whether plant species sampled near agricultural fields were more likely to contain viruses from plant species, or even compare viral host ranges (although Extended Data Figure 1 may speak to this). In addition, it is unclear if the authors intend to test if populations of plants growing near agriculture host more viruses than populations of those species that are far from agriculture; or if the plant species that grow in disturbed areas are more prone to viral spillover. Arguing for the first possibility would require additional sequencing of multiple plant populations, which is clearly outside of the scope of this paper. The second possibility would also require additional information and additional sequencing information of replicated agricultural habitats and controlling for plant species identity. In either case, I don't think that the current dataset can fully address this question.

Following from the point above, the conclusion in the discussion that “The identification of several plant traits that increase the plant-pollinator association, as well as the regions with high likelihood of pollen transfer between cultivated and wild plants as ‘hotspots’ of viral diversity, suggest that pollen-associated viruses could widely threaten plant biodiversity and food security.” seems entirely unfounded given the data. The study did not examine viral spillover or even quantify virus sharing between plant species in the main text so this should be removed unless it can be adequately supported by the data.

My second main comment is not a major flaw, but rather an observation that the pollen and floral traits included here are those that are already published and/or easy to measure. While this makes for an interesting comparison, it is not clear that these traits are necessarily related to pollen virus presence/diversity. Why not include other plant traits instead, potentially related to pollen development, or something about plant ecology or biology that is more relevant to viral infection or persistence? The discussion regarding this point (L 245-262) mainly addresses the potential for pollen-pollinator interactions and potential for viral dispersal/vectoring, but other pollen traits may have a more direct relationship to other aspects of viral life history. It would be helpful to expand discussion here.

Related to the question above:

Is among-population variation in viral incidence or diversity expected? If the authors' suggestion that disturbance and the potential for spillover is correct, then it may be that significant variation in viral incidence or diversity may exist within populations. If that is the case, it is unclear if plant species where viruses are not detected are either devoid of viruses or

if a population without viruses was sampled. Although the authors cannot address this using their own data (from Extended Data Table 3 it appears that a single population of each plant species was sampled for each species), this might be an important point to add to the discussion based on previous studies of more intensively sampled plant species.

A few more minor comments/questions:

Lines 70-71: what is a typical host range for pollen-borne viruses—do they typically infect a single host species or multiple? (or don't we know?)

Line 237-238: I see that the question above is addressed here: do you expect that closely related plants are more likely to be infected by the same virus?

Lines 66-75: the reason for the inclusion of the disturbance/agricultural gradient is not very clear from this section. What ecological processes do you expect to be driving patterns of viral diversity? Host density? Diversity of hosts? In combination with host range above, a bit more information on expectations here would be helpful.

The main questions mentioned in L 85-89 do not include the agricultural gradient information mentioned in the paragraph above, so again it is unclear why that is included in the introduction.

In addition, the figures in the supplementary material (particularly Extended Data Fig 1 and 3) and seem a bit more interesting than those currently presented in the main text as they present more information on detection among hosts, relatedness of plant hosts and relatedness of viruses.

Reviewer #3 (Remarks to the Author):

The manuscript titled “Proximity to human disturbance and flower traits shape the pollen virome” by the anonymous authors represents an interesting and undoubtedly novel research. It has, nevertheless, some serious flaws which, in my opinion, have to be addressed both conceptually and experimentally before the manuscript is considered for publication in Nature Comm. Nonetheless, I would encourage the authors to resubmit their manuscript after considerable improvement.

Firstly, I am not at all convinced that the approach chosen by the authors of this manuscript is entirely accurate: if their goal was “to uncover the diversity of pollen-associated viruses (for

the first time indeed), why would they disregard agricultural crops and only focus on a few selected wild species?

Also, I personally think that devoting so much attention to floral traits, pollen phenotypic plasticity, anthropogenic disturbances, levels of urbanization, geographic variations and other factors that might affect viral load in pollen, could be a second step of this interesting research, while initially it would be more reasonable to concentrate on the main subject that is, deciphering a comprehensive pollen virome in as many species as possible. This would let the authors to encompass the topic on a much larger scale.

This creates an impression that this research was initially intended to proceed towards more “botanical” course and then at some point authors developed interest in viruses and decided to add this topic to the project.

P3L48-52: It appears in the opening three sentences of the Introduction that pollinators are mainly responsible for transmission of plant viruses via pollen. Please note that although viruses can be present in pollen harbored by pollinators, it does not make them pollen-transmitted, meaning that they will infect the plant through the fertilized flower (horizontal transmission) or infect the seed and the seedling (vertical transmission).

P4L82: how exactly plants were determined to be “asymptomatic”? What range of symptoms the authors were looking at? Leaf yellowing, vein clearing, leaf distortions, growth abnormalities?

P6L126-128: ICTV has strict demarcation criteria for each taxonomic rank and classification of new viral species. The demarcation criteria vary but usually are much lower than 95% for different species. Anything within 90-95% percent of nucleotide or amino acid identity would likely correspond to a different strains or likely variants/isolates of the same viral species. Members of different (or novel species) would often have as low as 75% nt identity and less than ~ 80% aa identity in their selected ORFs (likely CP or Rep genes). Viruses have different thresholds: for potyviruses, for instance, a cut-off is less than 85% for a novel species. A threshold for mastreviruses of Geminiviridae is 75%. This of course varies among different families and genera. Its is quite strange therefore that the authors emerged with this 95% percent identity to classify novel viruses. This alone suggests that many of their "new viruses" are actually known species albeit there surely are some novel viruses found in the course of this research.

P6L136: just like several other viruses found in this study, this one is likely to be an incidental "passenger" virus, rather than a true part of the pollen virome - although it is probably valid to mention it under the virome “umbrella”.

P7L149: Extended Data Table 6. On the putative names assigned to the novel viruses in this

Table: only ICTV-accepted names of virus species are printed in italics and the first letters of the names are capitalized (please see ICTV rules for Orthography).

P7L150: at least one of these "novel" species appears to be a known mitovirus (LC006128.1), considering its query cover (100%) and 84 % identity. Although species demarcation criteria for mitoviruses have not been precisely defined, amino acid sequence identities of putative RdRp proteins between the different mitovirus species so far defined are less than 40%. (ICTV).

P7L150: The "novel" member of Bromoviridae family is most likely a strain of a well-known PNRSV (GeneBank IDs L38823.1, KT444702.1; JN416774.1, extended Data Table 6). Although specific levels of sequence similarity for ilarviruses have not been defined, for species of other genera in this family it is below 80%. 84-93% nucleotide identity most certainly indicate a different strain of PNRSV, rather than a novel virus.

P7L151: the same is true for the "novel" species belonging to Secoviridae. Coverage (99%) and percent identity (91-92%) indicate that this could be a strain of Tobacco ringspot nepovirus. Species Demarcation criteria for Secoviridae in conserved Pro-Pol region, acc. to ICTV, are less than 80% aa identity.

P7L149: Importantly, that claims of novelty for these and other viruses in the Extended Table 6 could not be verified independently because sequences are not available/supplied with the manuscript and the Bioproject number PRJNA589022 is not publicly available.

P7L159: many viruses in the Supplementary Table 1 could be known species as well, based on the query cover, percent identity and BLAST hits. Novelty of each of those viruses would have to be determined strictly in line with the ICTV criteria for the respective genera.

P8L180-181: members of these three families are long known to be transmitted vertically via seeds and pollen, meaning one does not need a randomization test to reveal that fact.

P9L188-189: distribution of viruses found in pollen is likely to be related to their host range. I have not noticed that authors specifically mentioned and/or discussed this important matter.

P9L201: this is most likely due to airborne bioparticles and fungal spores that are caught in spiky pollen surface patterns. It is however mentioned later in the text (P12L255).

P10L207: once again, viral richness has to be compared to a host range and occurrence of the respective host plants in those selected geographic areas.

P10L214: I would not call it intriguing: it is quite obvious that pollen grains of species that are in close proximity to anthropogenic environment would carry more contaminants potentially incorporating viral pathogens.

P10L217: Discussion. Based on the above comments, the discussion section would have to be completely restructured and rewritten.

P20L446: please define asymptomatic plants.

P20L449: if only perennial species were sampled, maybe the title of the paper should have been different.

P20L453: it is unlikely that all debris left after dismembrator could be removed by forceps. Therefore other sources of the described viruses cannot be excluded.

P28L623L: the bioproject PRJNA589022 is not publicly available to verify and evaluate all the sequences reported in the paper.

Responses to Reviewer #1

The manuscript entitled: “Proximity to human disturbance and flower traits shape the pollen virome” is a very well-written manuscript that further explores virus ecology. Specifically, this manuscript aimed at deciphering the diversity of pollen virome and further tried to disentangle landscape and floral factors that may have impacted the diversity and distribution of pollen viruses.

Thank you!

The major claims of the ms are that the pollen-associated virus diversity is much higher than previously thought and that both landscape (level of fragmentation by agriculture and urbanization) and plant traits (e.g. multiple flowered-inflorescences) are potentially playing a key-role in the taxonomic richness of the pollen virome and its geographic distribution.

While the conclusion of the study is original and interesting to others in the community, I have two major concerns about this manuscript:

1- Pollen purity. A specific protocol was developed for this study. Pollen was collected in situ using a sterile sonic dismembrator and non-pollen tissues (e.g., anther debris) were removed with sterile forceps. “Pure pollen” was then transferred to a collection tube with Lysing Matrix D that was kept on dry ice until being stored at -80°C at the University of Pittsburgh. I have concern about the purity of the pollen that was collected using this protocol, because only visible plant tissues were removed. The purity of pollen is crucial in this study while the authors are concluding that all viral contigs that were assembled were “pollen-associated”. This statement may actually be wrong because some of these viruses were potentially associated with plant debris (anther, leaf, etc.) that were probably contaminating the pollen fraction. To address this point, I would have suggested to add several controls. I’m not an expert of pollen purification but isolation and purification of highly purified, hydrated pollen grains can be achieved. Then, highly purifying a subset of the 145 plants that were analyzed in this study could have been carried out, for instance through flow cytometric sorting (e.g. Becker et al. (2003) *Plant Physiol.* 133, 713–725). In addition, I would have suggested to add plant controls, i.e. analyzing in parallel the plant virome of the plants from which the pollen was collected (again from a subset of the 145 plants that were analyzed in this study). This would have allowed distinguishing between viruses that were present in both pollen and plant viromes (and see whether they were phylogenetically related) and viruses that were only present in the pollen virome. The authors were aware about this problem of purity of the pollen while they have written in the discussion section “it is possible that the presence of the pollen-associated viruses is due to casual contact with other hosts that contacted the plants”.

We agree that having vegetative controls/tissue samples is a good idea, and would have allowed us to answer additional questions, but here we focused on identifying whether

there were viruses associated with pollen on a broad scale. This exploratory portion of the study is a needed first step to answer this question because no others have attempted to broadly characterize the pollen virome or related it to the ecology of plant hosts.

We agree pollen purity is important and took many precautions to safeguard for it. Although viruses on plant surfaces could have been dislodged mechanically via sonication, we would expect them to be in low abundance relative to true pollen-associated viruses. We looked for, but did not find, environmental viruses in our data set. We have noted this on Lines 333 – 339.

Nevertheless, we took the conservative stance of identifying viruses as “pollen-associated” instead of “pollen-transmitted” because we do not know the nature of their relationship to the pollen surveyed.

2- Influence of the region from which pollen was collected. While I think that this is convincing that there exists a previously unrecognized relationship between virus richness and plant traits, I have serious doubts as to whether the role of landscape (level of fragmentation by agriculture and urbanization) is relevant. Only one site (EDAFI) out of four was heavily fragmented by agriculture and urbanization. The EDAFI site represented only 18/145 samples (12.4%) that were processed. This site was compared to remote “wild” sites, which may have biased the comparison. I would have suggested to compare several neighboring “natural” and “disturbed” settings from the EDAFI region (and potentially repeat this design from other regions) to address this question of the role played by human-mediated disturbance on the diversity and distribution of the pollen virome.

We agree that our methods for quantifying the amount of disturbance in each landscape was coarse, and that our focus on the agricultural contribution to human land use did not acknowledge the autocorrelated changes, such as loss of natural vegetation. Thus, we have now standardized the area surrounding each host collection, calculated land use (the percent impervious surface, agriculture, and natural vegetation), and used an average to characterize each region.

Specifically, we used standardized radii around each collection site that capture both short- and long-distance pollinator foraging ranges (0.5 and 3 km). Land use within both radii was perfectly correlated ($r = 1$); thus, we present only the results for 3 km. We tested the relationship between the human land use (i.e., percent agriculture and impervious surface) and the conservative and relaxed estimates of virus richness using Spearman’s correlation tests (Lines 251 – 254).

We now use the term “land use” throughout the manuscript and acknowledge that many other attributes of a landscape beyond agriculture and impervious surface (e.g., the presence of invasive plant species and the loss of native vegetation) could lead to the association that we observed. Please see Lines 36 – 39, Lines 66 – 68, and Lines 251 – 254.

We also temper our conclusions about drivers of the landscape patterns in Lines 269 – 272. We agree that future studies could aim to pair “natural” and “disturbed” sites in each landscape to further elucidate the effect of land use on pollen virome richness.

Although 18 individual plants out of the 145 total plants were from EDAFI, our design strove to have *even sampling* of pollen at the species level. This did lead to different sampling of individuals per species (due to pollen production differences), which was accounted for statistically in the analyses and was a non-significant factor in virus richness (see Lines 614 – 623).

Responses to Reviewer #2

The manuscript “Proximity to human disturbance and flower traits shape the pollen virome” surveys plant species across landscape types to examine if plant species or environments are associated with the diversity viruses found in pollen. This topic is important and timely, and likely of very broad interest to pollination biologists, plant virologists and within ecology and evolution. Moreover, a broad survey of many plant species for viral presence in pollen has not been conducted previously so this study adds much to our understanding of this type of interaction.

Thank you!

The data presented here are novel, very interesting, and are surely to spur a lot of additional research on the topic. Congrats to the authors on this nice work. My main concern with the paper is that the conclusion regarding the link between disturbance/agriculture and virome structure is not adequately tested or supported. A few other methodological and statistical details are missing, as well as justification for the inclusion of the specific pollen traits included. As a result, I suggest a reframing of the paper to focus on its strength and what seems like its real contribution: describing the virome of a number of plant species which have not been previously examined. This contribution raises many ecological and evolutionary questions and will certainly spur lots of additional research on this topic. I detail the basis of my critiques below:

First, the data cannot address the main conclusion of the paper—that proximity to human disturbance and flower traits shape the pollen virome. This main message is also emphasized in the abstract (L 34-40) as a main finding of the paper. However, this conclusion is conceptually vague and not adequately tested. First, due to the sampling design used, the connection between human disturbance and viral composition or diversity cannot be directly made. This is because only a single sample from each plant species was included. Each of the plant species were sampled from a single bioregion, and as a result, plant species (and bioregion) is entirely confounded with human disturbance and so these potential predictors cannot be disentangled from each other. I could not find a description of the model related viral richness to human disturbance (Lines 214-215), so I do not know if plant species was accounted for in this model.

We agree with the reviewer concerning land use effects and have revised our approach to understanding land use effects on the pollen virome in response to Reviewer 1. In addition, we tempered our conclusions about human disturbance as a driver (relative to another correlated shift, like loss of natural vegetation) throughout (see above).

However, we use two levels of statistical control to address the questions of interest: 1) a nested design (where plant species are nested in their respective regions) and 2) explicit inclusion of phylogenetic relationships among plant species in our models. These allow us to treat species as replicates to ‘sample’ each of the four regions and to isolate the effects of specific plant traits of interest, controlling for the evolutionary history of the plant species. A full description of the models is found in Lines 683 – 695.

The second issue with this conclusion is that it is conceptually vague: the introduction states that wild plants growing in disturbed areas may be more prone to viral spillover from crops. However, the paper does not specifically address whether plant species sampled near agricultural fields were more likely to contain viruses from plant species, or even compare viral host ranges (although Extended Data Figure 1 may speak to this). In addition, it is unclear if the authors intend to test if populations of plants growing near agriculture host more viruses than populations of those species that are far from agriculture; or if the plant species that grow in disturbed areas are more prone to viral spillover. Arguing for the first possibility would require additional sequencing of multiple plant populations, which is clearly outside of the scope of this paper. The second possibility would also require additional information and additional sequencing information of replicated agricultural habitats and controlling for plant species identity. In either case, I don't think that the current dataset can fully address this question.

We agree that as originally written, the text was too brief and thus conceptually vague. We have revised the Introduction concerning land use and make more explicit the different mechanisms that can drive increased virus richness (see Lines 66 – 82). We also temper our conclusions concerning the direct mechanism that can be concluded from our study (see several places in the revised Discussion). Nevertheless, the work we present demonstrates that there is a signature of land use patterns (as a whole) on the pollen virome, which has not previously been shown.

Following from the point above, the conclusion in the discussion that “The identification of several plant traits that increase the plant-pollinator association, as well as the regions with high likelihood of pollen transfer between cultivated and wild plants as ‘hotspots’ of viral diversity, suggest that pollen-associated viruses could widely threaten plant biodiversity and food security.” seems entirely unfounded given the data. The study did not examine viral spillover or even quantify virus sharing between plant species in the main text so this should be removed unless it can be adequately supported by the data.

We agree that we did not examine viral spillover directly, but rather we identified 11 known viruses in more than one region or plant host, which at the least expands the known

viral host ranges of these viruses. We have revised the text to be more circumspect and make clear what is speculation from these findings in Lines 344 – 349 and Lines 353 – 355.

My second main comment is not a major flaw, but rather an observation that the pollen and floral traits included here are those that are already published and/or easy to measure. While this makes for an interesting comparison, it is not clear that these traits are necessarily related to pollen virus presence/diversity. Why not include other plant traits instead, potentially related to pollen development, or something about plant ecology or biology that is more relevant to viral infection or persistence? The discussion regarding this point (L 245-262) mainly addresses the potential for pollen-pollinator interactions and potential for viral dispersal/vectoring, but other pollen traits may have a more direct relationship to other aspects of viral life history. It would be helpful to expand discussion here.

We agree that additional plant or pollen grain traits may be related to the pollen virome and have noted this in Lines 318 – 320.

Related to the question above:

Is among-population variation in viral incidence or diversity expected? If the authors' suggestion that disturbance and the potential for spillover is correct, then it may be that significant variation in viral incidence or diversity may exist within populations. If that is the case, it is unclear if plant species where viruses are not detected are either devoid of viruses or if a population without viruses was sampled. Although the authors cannot address this using their own data (from Extended Data Table 3 it appears that a single population of each plant species was sampled for each species), this might be an important point to add to the discussion based on previous studies of more intensively sampled plant species.

We agree, and we have expanded the discussion of within population variation in Lines 287 – 292 and in Lines 350 – 353.

A few more minor comments/questions:

Lines 70-71: what is a typical host range for pollen-borne viruses—do they typically infect a single host species or multiple? (or don't we know?)

This is a good question and is one for which there is no 'typical' answer. Our work is a step in the direction of addressing this big knowledge gap, especially for pollen-associated viruses.

Line 237-238: I see that the question above is addressed here: do you expect that closely related plants are more likely to be infected by the same virus?

This is also not known for pollen-associated viruses. For some, this could be the case; perhaps for others, and perhaps even for other viruses (i.e., SARS-CoV-2), maybe not.

Lines 66-75: the reason for the inclusion of the disturbance/agricultural gradient is not very clear from this section. What ecological processes do you expect to be driving patterns of viral diversity? Host density? Diversity of hosts? In combination with host range above, a bit more information on expectations here would be helpful.

This section of the Introduction has been expanded upon, see comment above (and see Lines 66 – 82).

The main questions mentioned in L 85-89 do not include the agricultural gradient information mentioned in the paragraph above, so again it is unclear why that is included in the introduction.

This section of the Introduction has been revised for clarity (see Line 94).

In addition, the figures in the supplementary material (particularly Extended Data Fig 1 and 3) and seem a bit more interesting than those currently presented in the main text as they present more information on detection among hosts, relatedness of plant hosts and relatedness of viruses.

We agree that the original Extended Data Figures 1 and 3 give the reader helpful visual representation of some of our main points, so they are now included as Figures 3 and 4.

Responses to Reviewer #3

The manuscript titled “Proximity to human disturbance and flower traits shape the pollen virome” by the anonymous authors represents an interesting and undoubtedly novel research. It has, nevertheless, some serious flaws which, in my opinion, have to be addressed both conceptually and experimentally before the manuscript is considered for publication in Nature Comm. Nonetheless, I would encourage the authors to resubmit their manuscript after considerable improvement.

Thank you!

Firstly, I am not at all convinced that the approach chosen by the authors of this manuscript is entirely accurate: if their goal was “to uncover the diversity of pollen-associated viruses (for the first time indeed), why would they disregard agricultural crops and only focus on a few selected wild species?

Nearly all studies on pollen-associated viruses have focused on crop or cultivated plant hosts, leaving the pollen virome of wild plant species mostly uncharacterized. Therefore, we chose to focus completely on a diverse assemblage of wild plant species instead of agricultural crops in the current study.

We have changed the title to “Land use and floral traits shape the pollen virome of wild plants” to make our focus more clear. We highlight the need to characterize the pollen virome of wild plants in the Introduction in Lines 59 – 65.

Also, I personally think that devoting so much attention to floral traits, pollen phenotypic plasticity, anthropogenic disturbances, levels of urbanization, geographic variations and other factors that might affect viral load in pollen, could be a second step of this interesting research, while initially it would be more reasonable to concentrate on the main subject that is, deciphering a comprehensive pollen virome in as many species as possible. This would let the authors to encompass the topic on a much larger scale.

This creates an impression that this research was initially intended to proceed towards more “botanical” course and then at some point authors developed interest in viruses and decided to add this topic to the project.

We agree that characterizing the pollen virome of as many plant species as possible, especially wild ones, would be ideal. With the current study, we hope that we have set the stage for a large body of future work that includes deciphering the pollen virome of many more plant species.

P3L48-52: It appears in the opening three sentences of the Introduction that pollinators are mainly responsible for transmission of plant viruses via pollen. Please note that although viruses can be present in pollen harbored by pollinators, it does not make them pollen-transmitted, meaning that they will infect the plant through the fertilized flower (horizontal transmission) or infect the seed and the seedling (vertical transmission).

You are correct, thank you for this reminder. It is for these reasons, and because we did not know whether the identified viruses were on the outside or the inside of the sampled pollen grains, that we call the viruses that we identified “pollen-associated” and not “pollen-transmitted” throughout the manuscript.

P4L82: how exactly plants were determined to be “asymptomatic”? What range of symptoms the authors were looking at? Leaf yellowing, vein clearing, leaf distortions, growth abnormalities?

None of the sampled plants displayed classic virus symptoms, and we now state this in Lines 519 – 521. In addition, we now refer to the sampled plants as “visually asymptomatic” throughout the manuscript.

P6L126-128: ICTV has strict demarcation criteria for each taxonomic rank and classification of new viral species. The demarcation criteria vary but usually are much lower than 95% for different species. Anything within 90-95% percent of nucleotide or amino acid identity would likely correspond to a different strains or likely variants/isolates of the same viral species. Members of different (or novel species) would often have as low as 75% nt identity and less than ~ 80% aa identity in their selected ORFs (likely CP or Rep genes). Viruses have different thresholds: for potyviruses, for instance, a cut-off is less than 85% for a novel species. A threshold for mastreviruses of Geminiviridae is 75%. This of course varies among different

families and genera. It is quite strange therefore that the authors emerged with this 95% percent identity to classify novel viruses. This alone suggests that many of their "new viruses" are actually known species albeit there surely are some novel viruses found in the course of this research.

We now use the ICTV family-specific percent identity demarcation criteria to distinguish between novel coding-complete strains of known viruses, novel coding-complete viral genomes (new Extended Data Table 5), novel partial strains of known viruses, and novel partial viral genomes (Supplementary Table 1). The previous Extended Data Table 5 was revised to identify the partial novel strains, but they are now included in and indicated as such in Supplementary Table 1.

Novelty (strain vs. genome) in each represented viral family was made based upon the following criteria from the ICTV, and these thresholds are noted in Extended Data Table 5 and Supplementary Table 1:

- *Bromoviridae, Aspiviridae, Chrysoviridae, Idaeovirus, "No Family", Phenuiviridae, Rhabdoviridae, Solemoviridae, Endornaviridae, Mononegavirales*: <80% identity between nucleotide sequences
- *Partitiviridae*: <90% identity between amino acid sequences of the RdRp or <80% identity between amino acid sequences of the coat protein
- *Secoviridae*: <75% identity between amino acid sequences of the coat protein or <80% identity between amino acid sequences of the Pro-Pol polyprotein (i.e., RdRp or others on the same segment)
- *Amalgaviridae*: <75% identity between amino acid sequences
- *Narnaviridae*: <40 – 50% identity between amino acid sequences
- *Alphaflexiviridae, Betaflexiviridae*: <72% identity between nucleotide sequences of the coat protein or RdRp
- *Caulimoviridae*: <80% identity between nucleotide sequences
- *Geminiviridae*: <75% identity between nucleotide sequences
- *Iflaviridae, Luteoviridae, Peribunyaviridae*: <90% identity between amino acid sequences
- *Ourmiavirus (Botourmiaviridae)*: <70% identity between amino acid sequences
- *Tombusviridae*: <85% identity between amino acid sequences.

We have revised the Results (see Lines 141 – 145) and the Methods (Lines 583 – 586) to reflect this change in classification criteria.

Using the family-specific classification criteria of the ICTV did not change our overall conclusions and only slightly changed our counts of known viruses (Extended Data Table 4), novel coding-complete genomes and strains of known viruses (Extended Data Table 5), and novel partial viral genomes and strains of known viruses (Supplementary Table 1). These modest changes have been updated in the text in the Abstract/Summary (Lines 32 –

33), the Results (Lines 152 – 201), and the Discussion (Lines 262 – 266). In addition, we reran all statistical analyses and updated the appropriate text in the Results (Lines 210 – 254) and the Methods (Lines 564 and 614 – 620). It is important to note that while many of the reported values changed, the interpretation of the results remains the same, as the changes were very slight.

P6L136: just like several other viruses found in this study, this one is likely to be an incidental "passenger" virus, rather than a true part of the pollen virome - although it is probably valid to mention it under the virome "umbrella".

While it is true that *Deformed wing virus* is a bee pathogen and may be a "passenger" virus, it is known to be transmitted between colonies via infected pollen. In addition to being on the outside of pollen grains, it may also be tightly bound to their outer layer.

We have revised the sentence in question and added another immediately afterward (Lines 157 – 160) to better explain its presence. We have also made this point more clear in the Discussion (Lines 328 – 332).

P7L149: Extended Data Table 6. On the putative names assigned to the novel viruses in this Table: only ICTV-accepted names of virus species are printed in italics and the first letters of the names are capitalized (please see ICTV rules for Orthography).

Thank you, we have un-italicized all names of our novel viruses, both in Extended Data Table 5 (formerly Extended Data Table 6) and Supplementary Table 1.

P7L150: at least one of these "novel" species appears to be a known mitovirus (LC006128.1), considering its query cover (100%) and 84 % identity. Although species demarcation criteria for mitoviruses have not been precisely defined, amino acid sequence identities of putative RdRp proteins between the different mitovirus species so far defined are less than 40%. (ICTV).

We agree that the previous *Solidago narnavirus 1* is a novel strain of *Fusarium globosum mitovirus 1* based upon ICTV demarcation criteria for the *Narnaviridae* viral family. In Extended Data Table 5 and elsewhere, this virus is now referred to *Fusarium globosum mitovirus 1* (novel strain 1). This change has also been highlighted in Extended Data Table 5.

P7L150: The "novel" member of Bromoviridae family is most likely a strain of a well-known PNRSV (GeneBank IDs L38823.1, KT444702.1; JN416774.1, extended Data Table 6). Although specific levels of sequence similarity for ilarviruses have not been defined, for species of other genera in this family it is below 80%. 84-93% nucleotide identity most certainly indicate a different strain of PNRSV, rather than a novel virus.

We agree that the previous *Packera aurea* bromovirus 1 is a novel strain of *Prunus necrotic ringspot virus* based upon ICTV demarcation criteria for the *Bromoviridae* viral family. In Extended Data Table 5 and elsewhere, this virus is now referred to *Prunus necrotic ringspot virus* (novel strain 1). This change has also been highlighted in Extended Data Table 5.

P7L151: the same is true for the "novel" species belonging to Secoviridae. Coverage (99%) and percent identity (91-92%) indicate that this could be a strain of Tobacco ringspot nepovirus. Species Demarcation criteria for Secoviridae in conserved Pro-Pol region, acc. to ICTV, are less than 80% aa identity.

We agree that the previous *Oenothera biennis* secovirus 1 is a novel strain of *Tobacco ringspot virus* based upon ICTV demarcation criteria for the *Secoviridae* viral family. In Extended Data Table 5 and elsewhere, this virus is now referred to *Tobacco ringspot virus* (novel strain 1). This change has also been highlighted in Extended Data Table 5.

P7L149: Importantly, that claims of novelty for these and other viruses in the Extended Table 6 could not be verified independently because sequences are not available/supplied with the manuscript and the Bioproject number PRJNA589022 is not publicly available.

Please see our response to your last comment below regarding Bioproject number PRJNA589022 and availability of sequences from the current study.

P7L159: many viruses in the Supplementary Table 1 could be known species as well, based on the query cover, percent identity and BLAST hits. Novelty of each of those viruses would have to be determined strictly in line with the ICTV criteria for the respective genera.

Using the above family-specific ICTV demarcation criteria, we re-evaluated the status of all the novel partial genomes listed in Supplementary Table 1. With these new criteria, some are now designated as a novel strain of a known virus, but remain in the table. In addition, we added the novel partial strains of the known viruses that were originally listed in the previous Extended Data Table 5 to Supplementary Table 1, for the same reasons. All revisions and additions have been highlighted in Supplementary Table 1.

P8L180-181: members of these three families are long known to be transmitted vertically via seeds and pollen, meaning one does not need a randomization test to reveal that fact.

We wanted to be able to confirm our observations with statistics. Therefore, we think that we should still report the results of the randomization test, especially since not all viruses belonging to these three families may be vertically transmitted via seeds and pollen, and since we discovered novel viruses and strains representing numerous other viral families.

P9L188-189: distribution of viruses found in pollen is likely to be related to their host range. I have not noticed that authors specifically mentioned and/or discussed this important matter.

We think that you raise an excellent point. We have now mentioned this possibility twice in the Discussion, in Lines 282 – 286 and in Lines 322 – 324. The latter builds upon a hypothesis in the sentence immediately before.

P9L201: this is most likely due to airborne bioparticles and fungal spores that are caught in spiky pollen surface patterns. It is however mentioned later in the text (P12L255).

This is an interesting point. We have added a sentence to the Discussion (Lines 312 – 313) as another reason why plants with spiky pollen grains may have more pollen-associated viruses than those with smoother pollen grains.

P10L207: once again, viral richness has to be compared to a host range and occurrence of the respective host plants in those selected geographic areas.

We agree that host range likely plays an important role in structuring the pollen virome, but the full wild plant host range for many plant viruses, including pollen-associated ones, is not known. Characterizing the pollen virome of wild plant species is the first step in addressing this gap in plant virus ecology so that future studies can explore more explicit hypotheses concerning host range, host presence, landscape attributes, and pollen virome richness.

P10L214: I would not call it intriguing: it is quite obvious that pollen grains of species that are in close proximity to anthropogenic environment would carry more contaminants potentially incorporating viral pathogens.

This has been noted, and we have removed the word “intriguing” from the sentence in question.

P10L217: Discussion. Based on the above comments, the discussion section would have to be completely restructured and rewritten.

We agree, thank you for your comments. The Discussion has been greatly revised.

P20L446: please define asymptomatic plants.

Please see our above response to your comment concerning P4, L82 in the Introduction.

P20L449: if only perennial species were sampled, maybe the title of the paper should have been different.

While most of the surveyed plant species were perennials, three were not. Thus, we do not think that we should further revise our title from “Land use and floral traits shape the

pollen virome of wild plants”, following one of your earlier suggestions. We have also revised the sentence in question (Lines 523 – 524) to make this more clear.

P20L453: it is unlikely that all debris left after dismembrator could be removed by forceps. Therefore other sources of the described viruses cannot be excluded.

You are correct. We removed visible anther/plant debris and thrips with sterile forceps, and though it is possible that microscopic debris remained, we did not detect common environmental contaminants in any of our samples. We noted this in the Discussion in response to Reviewer 1 (Lines 333 – 339).

In the Methods, we have revised the next sentence after the one in question (Lines 528 – 531).

P28L623L: the bioproject PRJNA589022 is not publicly available to verify and evaluate all the sequences reported in the paper.

The raw reads are set to be publicly available in GenBank under Bioproject number PRJNA589022 upon publication of the current study. In the meantime, the contig sequences are available in Supplementary Dataset 2.

This has been made more clear in our “Data availability” statement (Lines 716 – 722).

Thank you again for your comments and for those of the reviewers. We have fully addressed all of them, including those concerning the purity of our pollen samples, and you will notice that, as a result, the paper is even stronger than our first resubmission. Please find below a summary of the changes made in response to the reviews (in **bold**). The line numbers refer to the track-changes version of the manuscript that we have uploaded. Changes made to the tables and supplementary documents are either highlighted in yellow or are in track-changes in the revised versions.

Sincerely,
Andrea Fetters and Tia-Lynn Ashman, for all authors.

Responses to Reviewer #1

The manuscript entitled: “Proximity to human disturbance and flower traits shape the pollen virome” is a very well-written manuscript that further explores virus ecology. Specifically, this manuscript aimed at deciphering the diversity of pollen virome and further tried to disentangle landscape and floral factors that may have impacted the diversity and distribution of pollen viruses.

Thank you!

The major claims of the ms are that the pollen-associated virus diversity is much higher than previously thought and that both landscape (level of fragmentation by agriculture and urbanization) and plant traits (e.g. multiple flowered-inflorescences) are potentially playing a key-role in the taxonomic richness of the pollen virome and its geographic distribution.

While the conclusion of the study is original and interesting to others in the community, I have two major concerns about this manuscript:

1- Pollen purity. A specific protocol was developed for this study. Pollen was collected in situ using a sterile sonic dismembrator and non-pollen tissues (e.g., anther debris) were removed with sterile forceps. “Pure pollen” was then transferred to a collection tube with Lysing Matrix D that was kept on dry ice until being stored at -80°C at the University of Pittsburgh. I have concern about the purity of the pollen that was collected using this protocol, because only visible plant tissues were removed. The purity of pollen is crucial in this study while the authors are concluding that all viral contigs that were assembled were “pollen-associated”. This statement may actually be wrong because some of these viruses were potentially associated with plant debris (anther, leaf, etc.) that were probably contaminating the pollen fraction. To address this point, I would have suggested to add several controls. I’m not an expert of pollen purification but isolation and purification of highly purified, hydrated pollen grains can be achieved. Then, highly purifying a subset of the 145 plants that were analyzed in this study could have been carried out, for instance through flow cytometric sorting (e.g. Becker et al. (2003) Plant Physiol.

133, 713–725). In addition, I would have suggested to add plant controls, i.e. analyzing in parallel the plant virome of the plants from which the pollen was collected (again from a subset of the 145 plants that were analyzed in this study). This would have allowed distinguishing between viruses that were present in both pollen and plant viromes (and see whether they were phylogenetically related) and viruses that were only present in the pollen virome. The authors were aware about this problem of purity of the pollen while they have written in the discussion section “it is possible that the presence of the pollen-associated viruses is due to casual contact with other hosts that contacted the plants”.

We agree that having vegetative controls/tissue samples is a good idea, and they would have allowed us to answer additional questions, but here we focused on identifying whether there were viruses associated with pollen on a broad scale. The exploratory portion of the study is a needed first step to answer this question because no others have attempted to broadly characterize the pollen virome or relate it to the ecology of the plant hosts.

We agree pollen sample purity is important and took many precautions to safeguard for it. Although viruses on plant surfaces could have been dislodged mechanically via sonication, we would expect them to be in low abundance relative to true pollen-associated viruses. We looked for, but did not find, environmental viruses in our data set. We have noted this in Lines 338 – 344.

In addition, we now include a Supplementary Methods file, three Supplementary Methods tables, and a Supplementary Methods figure wherein we detail the evaluations of pollen sample purity that we performed. In the main text, we point the reader to these files in Lines 533 – 536. Briefly, we evaluated pollen sample purity in two ways. First, we assessed the level of physical contamination in pollen samples from three surveyed plant species using light microscopy. Second, we evaluated the potential for vegetative contamination using the expression of chloroplast-specific genes as an indicator given that in the majority of plant species, chloroplasts are almost exclusively maternally inherited and are thus not incorporated in pollen grains. We evaluated the relative expression levels of chloroplast-specific genes to pollen-specific genes in RNAseq analyses (two surveyed plant species) and a RT-PCR experiment (one surveyed plant species).

The microscopic analysis revealed very little physical contamination in our pollen samples, and pollen purity estimates ranged from 99.6 – 99.8%. The RNAseq analyses and the RT-PCR experiment both demonstrated the near-absence of vegetative tissue in the pollen samples as we found that the relative expression of chloroplast-specific genes was high in leaves and nearly non-detectable in pollen.

Nevertheless, we took the conservative stance of identifying viruses as “pollen-associated” instead of “pollen-transmitted” because we do not know the nature of their relationship to the pollen surveyed.

2- Influence of the region from which pollen was collected. While I think that this is convincing that there exists a previously unrecognized relationship between virus richness and plant traits, I have serious doubts as to whether the role of landscape (level of fragmentation by agriculture and urbanization) is relevant. Only one site (EDAFI) out of four was heavily fragmented by agriculture and urbanization. The EDAFI site represented only 18/145 samples (12.4%) that were processed. This site was compared to remote “wild” sites, which may have biased the comparison. I would have suggested to compare several neighboring “natural” and “disturbed” settings from the EDAFI region (and potentially repeat this design from other regions) to address this question of the role played by human-mediated disturbance on the diversity and distribution of the pollen virome.

We agree that our methods for quantifying the amount of disturbance in each landscape was coarse and that our focus on the agricultural contribution to human land use did not acknowledge the autocorrelated changes, such as loss of natural vegetation. Thus, we have now standardized the area surrounding each pollen sample collection, calculated land use (the percent impervious surface, agriculture, and natural vegetation), and used an average to characterize each region.

Specifically, we used standardized radii around each collection site that capture both short- and long-distance pollinator foraging ranges (0.5 and 3 km). Land use within both radii was perfectly correlated ($r = 1$); thus, we present only the results for 3 km. We tested the relationship between the human land use (i.e., percent agriculture and impervious surface) and the conservative and relaxed estimates of virus richness using Spearman’s correlation tests (Lines 256 – 259).

We now use the term “land use” throughout the manuscript and acknowledge that many other attributes of a landscape beyond agriculture and impervious surface (e.g., the presence of invasive plant species and the loss of native vegetation) could lead to the association that we observed. Please see Lines 36 – 39, Lines 66 – 68, and Lines 256 – 259.

We also temper our conclusions about the drivers of the landscape patterns in Lines 274 – 277. We agree that future studies could aim to pair “natural” and “disturbed” sites in each landscape to further elucidate the effect of land use on pollen virome richness.

Although 18 individual plants out of the 145 total sampled plants were from EDAFI, our design strove to have even sampling of pollen at the species level. This did lead to different sampling of individuals per species (due to pollen production differences), which was accounted for statistically in the analyses and was a non-significant factor in virus richness (see Lines 622 – 631).

Responses to Reviewer #2

The manuscript “Proximity to human disturbance and flower traits shape the pollen virome” surveys plant species across landscape types to examine if plant species or environments are

associated with the diversity viruses found in pollen. This topic is important and timely, and likely of very broad interest to pollination biologists, plant virologists and within ecology and evolution. Moreover, a broad survey of many plant species for viral presence in pollen has not been conducted previously so this study adds much to our understanding of this type of interaction.

Thank you!

The data presented here are novel, very interesting, and are surely to spur a lot of additional research on the topic. Congrats to the authors on this nice work. My main concern with the paper is that the conclusion regarding the link between disturbance/agriculture and virome structure is not adequately tested or supported. A few other methodological and statistical details are missing, as well as justification for the inclusion of the specific pollen traits included. As a result, I suggest a reframing of the paper to focus on its strength and what seems like its real contribution: describing the virome of a number of plant species which have not been previously examined. This contribution raises many ecological and evolutionary questions and will certainly spur lots of additional research on this topic. I detail the basis of my critiques below:

First, the data cannot address the main conclusion of the paper—that proximity to human disturbance and flower traits shape the pollen virome. This main message is also emphasized in the abstract (L 34-40) as a main finding of the paper. However, this conclusion is conceptually vague and not adequately tested. First, due to the sampling design used, the connection between human disturbance and viral composition or diversity cannot be directly made. This is because only a single sample from each plant species was included. Each of the plant species were sampled from a single bioregion, and as a result, plant species (and bioregion) is entirely confounded with human disturbance and so these potential predictors cannot be disentangled from each other. I could not find a description of the model related viral richness to human disturbance (Lines 214-215), so I do not know if plant species was accounted for in this model.

We agree with the reviewer concerning land use effects and have revised our approach to understanding land use effects on the pollen virome in response to Reviewer 1. In addition, we tempered our conclusions about human disturbance as a driver (relative to another correlated shift, like loss of natural vegetation) throughout (see above).

However, we use two levels of statistical control to address the questions of interest: 1) a nested design (where plant species are nested in their respective regions) and 2) explicit inclusion of phylogenetic relationships among plant species in our models. These allow us to treat species as replicates to ‘sample’ each of the four regions and to isolate the effects of specific plant traits of interest, controlling for the evolutionary history of the plant species. A full description of the models is found in Lines 691 – 703.

The second issue with this conclusion is that it is conceptually vague: the introduction states that wild plants growing in disturbed areas may be more prone to viral spillover from crops.

However, the paper does not specifically address whether plant species sampled near agricultural fields were more likely to contain viruses from plant species, or even compare viral host ranges (although Extended Data Figure 1 may speak to this). In addition, it is unclear if the authors intend to test if populations of plants growing near agriculture host more viruses than populations of those species that are far from agriculture; or if the plant species that grow in disturbed areas are more prone to viral spillover. Arguing for the first possibility would require additional sequencing of multiple plant populations, which is clearly outside of the scope of this paper. The second possibility would also require additional information and additional sequencing information of replicated agricultural habitats and controlling for plant species identity. In either case, I don't think that the current dataset can fully address this question.

We agree that as originally written, the text was too brief and thus conceptually vague. We have revised the Introduction concerning land use and make more explicit the different mechanisms that can drive increased virus richness (see Lines 66 – 82). We also temper our conclusions concerning the direct mechanism that can be concluded from our study (see several places in the revised Discussion). Nevertheless, the work we present demonstrates that there is a signature of land use patterns (as a whole) on the pollen virome, which has not previously been shown.

Following from the point above, the conclusion in the discussion that “The identification of several plant traits that increase the plant-pollinator association, as well as the regions with high likelihood of pollen transfer between cultivated and wild plants as ‘hotspots’ of viral diversity, suggest that pollen-associated viruses could widely threaten plant biodiversity and food security.” seems entirely unfounded given the data. The study did not examine viral spillover or even quantify virus sharing between plant species in the main text so this should be removed unless it can be adequately supported by the data.

We agree that we did not examine viral spillover directly, but rather we identified 10 known viruses in more than one region or plant species, which at the least expands the known viral host ranges of these viruses. We have revised the text to be more circumspect and make clear what is speculation from these findings in Lines 349 – 354 and Lines 358 – 360.

My second main comment is not a major flaw, but rather an observation that the pollen and floral traits included here are those that are already published and/or easy to measure. While this makes for an interesting comparison, it is not clear that these traits are necessarily related to pollen virus presence/diversity. Why not include other plant traits instead, potentially related to pollen development, or something about plant ecology or biology that is more relevant to viral infection or persistence? The discussion regarding this point (L 245-262) mainly addresses the potential for pollen-pollinator interactions and potential for viral dispersal/vectoring, but other pollen traits may have a more direct relationship to other aspects of viral life history. It would be helpful to expand discussion here.

We agree that additional plant or pollen grain traits may be related to the pollen virome and have noted this in Lines 323 – 325.

Related to the question above:

Is among-population variation in viral incidence or diversity expected? If the authors' suggestion that disturbance and the potential for spillover is correct, then it may be that significant variation in viral incidence or diversity may exist within populations. If that is the case, it is unclear if plant species where viruses are not detected are either devoid of viruses or if a population without viruses was sampled. Although the authors cannot address this using their own data (from Extended Data Table 3 it appears that a single population of each plant species was sampled for each species), this might be an important point to add to the discussion based on previous studies of more intensively sampled plant species.

We agree, and we have expanded the discussion of within population variation in Lines 292 – 297 and in Lines 355 – 358.

A few more minor comments/questions:

Lines 70-71: what is a typical host range for pollen-borne viruses—do they typically infect a single host species or multiple? (or don't we know?)

This is a good question and is one for which there is no 'typical' answer. Our work is a step in the direction of addressing this big knowledge gap, especially for pollen-associated viruses.

Line 237-238: I see that the question above is addressed here: do you expect that closely related plants are more likely to be infected by the same virus?

This is also not known for pollen-associated viruses. For some, this could be the case; perhaps for others, and perhaps even for viruses infecting other types of hosts (i.e., SARS-CoV-2), maybe not.

Lines 66-75: the reason for the inclusion of the disturbance/agricultural gradient is not very clear from this section. What ecological processes do you expect to be driving patterns of viral diversity? Host density? Diversity of hosts? In combination with host range above, a bit more information on expectations here would be helpful.

This section of the Introduction has been expanded upon, see comment above (and see Lines 66 – 82).

The main questions mentioned in L 85-89 do not include the agricultural gradient information mentioned in the paragraph above, so again it is unclear why that is included in the introduction.

This section of the Introduction has been revised for clarity (see Lines 94 – 95).

In addition, the figures in the supplementary material (particularly Extended Data Fig 1 and 3) and seem a bit more interesting than those currently presented in the main text as they present more information on detection among hosts, relatedness of plant hosts and relatedness of viruses.

We agree that the original Extended Data Figures 1 and 3 give the reader helpful visual representation of some of our main points, so they are now included as Figures 3 and 4.

Responses to Reviewer #3

The manuscript titled “Proximity to human disturbance and flower traits shape the pollen virome” by the anonymous authors represents an interesting and undoubtedly novel research. It has, nevertheless, some serious flaws which, in my opinion, have to be addressed both conceptually and experimentally before the manuscript is considered for publication in Nature Comm. Nonetheless, I would encourage the authors to resubmit their manuscript after considerable improvement.

Thank you!

Firstly, I am not at all convinced that the approach chosen by the authors of this manuscript is entirely accurate: if their goal was “to uncover the diversity of pollen-associated viruses (for the first time indeed), why would they disregard agricultural crops and only focus on a few selected wild species?

Nearly all studies on pollen-associated viruses have focused on crop or cultivated plant hosts, leaving the pollen virome of wild plant species mostly uncharacterized. Therefore, we chose to focus completely on a diverse assemblage of wild plant species instead of agricultural crops in the current study.

We have changed the title to “Land use and floral traits shape the pollen virome of wild plants” to make our focus more clear. We highlight the need to characterize the pollen virome of wild plants in the Introduction in Lines 59 – 65.

Also, I personally think that devoting so much attention to floral traits, pollen phenotypic plasticity, anthropogenic disturbances, levels of urbanization, geographic variations and other factors that might affect viral load in pollen, could be a second step of this interesting research, while initially it would be more reasonable to concentrate on the main subject that is, deciphering a comprehensive pollen virome in as many species as possible. This would let the authors to encompass the topic on a much larger scale.

This creates an impression that this research was initially intended to proceed towards more “botanical” course and then at some point authors developed interest in viruses and decided to add this topic to the project.

We agree that characterizing the pollen virome of as many plant species as possible, especially wild ones, would be ideal. With the current study, we hope that we have set the stage for a large body of future work that includes deciphering the pollen virome of many more plant species.

P3L48-52: It appears in the opening three sentences of the Introduction that pollinators are mainly responsible for transmission of plant viruses via pollen. Please note that although viruses can be present in pollen harbored by pollinators, it does not make them pollen-transmitted, meaning that they will infect the plant through the fertilized flower (horizontal transmission) or infect the seed and the seedling (vertical transmission).

You are correct, thank you for this reminder. It is for these reasons, and because we did not know whether the identified viruses were on the outside or the inside of the sampled pollen grains, that we call the viruses that we identified “pollen-associated” and not “pollen-transmitted” throughout the manuscript.

P4L82: how exactly plants were determined to be “asymptomatic? What range of symptoms the authors were looking at? Leaf yellowing, vein clearing, leaf distortions, growth abnormalities?

None of the sampled plants displayed classic virus symptoms, and we now state this in Lines 524 – 526. In addition, we now refer to the sampled plants as “visually asymptomatic” throughout the manuscript.

P6L126-128: ICTV has strict demarcation criteria for each taxonomic rank and classification of new viral species. The demarcation criteria vary but usually are much lower than 95% for different species. Anything within 90-95% percent of nucleotide or amino acid identity would likely correspond to a different strains or likely variants/isolates of the same viral species. Members of different (or novel species) would often have as low as 75% nt identity and less than ~ 80% aa identity in their selected ORFs (likely CP or Rep genes). Viruses have different thresholds: for potyviruses, for instance, a cut-off is less than 85% for a novel species. A threshold for mastreviruses of Geminiviridae is 75%. This of course varies among different families and genera. Its is quite strange therefore that the authors emerged with this 95% percent identity to classify novel viruses. This alone suggests that many of their "new viruses" are actually known species albeit there surely are some novel viruses found in the course of this research.

We now use the ICTV family-specific percent identity demarcation criteria to distinguish between novel coding-complete strains of known viruses, novel coding-complete viral genomes (new Extended Data Table 5), novel partial strains of known viruses, and novel partial viral genomes (Supplementary Table 1). The previous Extended Data Table 5 was revised to identify the partial novel strains, but they are now included in and indicated as such in Supplementary Table 1.

Novelty (strain vs. genome) in each represented viral family was made based upon the following criteria from the ICTV, and these thresholds are noted in Extended Data Table 5 and Supplementary Table 1:

- **Bromoviridae, Aspiviridae, Chrysoviridae, *Idaeovirus* (Mayoviridae), “No Family”, Phenuiviridae, Rhabdoviridae, Solemoviridae, Endornaviridae, Mononegavirales: <80% identity between nucleotide sequences**
- **Partitiviridae: <90% identity between amino acid sequences of the RdRp or <80% identity between amino acid sequences of the coat protein**
- **Secoviridae: <75% identity between amino acid sequences of the coat protein or <80% identity between amino acid sequences of the Pro-Pol polyprotein (i.e., RdRp or others on the same segment)**
- **Amalgaviridae: <75% identity between amino acid sequences**
- **Narnaviridae: <40 – 50% identity between amino acid sequences**
- **Alphaflexiviridae, Betaflexiviridae: <72% identity between nucleotide sequences of the coat protein or RdRp**
- **Caulimoviridae: <80% identity between nucleotide sequences**
- **Geminiviridae: <75% identity between nucleotide sequences**
- **Iflaviridae, Luteoviridae, Peribunyaviridae: <90% identity between amino acid sequences**
- ***Ourmiavirus* (Botourmiaviridae): <70% identity between amino acid sequences**
- **Tombusviridae: <85% identity between amino acid sequences.**

We have revised the Results (see Lines 142 – 146) and the Methods (Lines 590 – 594) to reflect this change in classification criteria.

Using the family-specific classification criteria of the ICTV did not change our overall conclusions and only slightly changed our counts of known viruses (Extended Data Table 4), novel coding-complete genomes and strains of known viruses (Extended Data Table 5), and novel partial viral genomes and strains of known viruses (Supplementary Table 1). These modest changes have been updated in the text in the Abstract/Summary (Lines 32 – 34), the Results (Lines 154 – 206), and the Discussion (Lines 267 – 271). In addition, we reran all statistical analyses and updated the appropriate text in the Results (Lines 215 – 259) and the Methods (Lines 572 – 573 and 622 – 631). It is important to note that while many of the reported values changed, the interpretation of the results remains the same, as the changes were very slight.

P6L136: just like several other viruses found in this study, this one is likely to be an incidental "passenger" virus, rather than a true part of the pollen virome - although it is probably valid to mention it under the virome “umbrella”.

While it is true that *Deformed wing virus* is a bee pathogen and may be a “passenger” virus, it is known to be transmitted between colonies via infected pollen. In addition to being on the outside of pollen grains, it may also be tightly bound to their outer layer.

We have revised the sentence in question and added another immediately afterward (Lines 159 – 162) to better explain its presence. We have also made this point more clear in the Discussion (Lines 333 – 337).

P7L149: Extended Data Table 6. On the putative names assigned to the novel viruses in this Table: only ICTV-accepted names of virus species are printed in italics and the first letters of the names are capitalized (please see ICTV rules for Orthography).

Thank you, we have un-italicized all names of our novel viruses, both in Extended Data Table 5 (formerly Extended Data Table 6) and Supplementary Table 1.

P7L150: at least one of these "novel" species appears to be a known mitovirus (LC006128.1), considering its query cover (100%) and 84 % identity. Although species demarcation criteria for mitoviruses have not been precisely defined, amino acid sequence identities of putative RdRp proteins between the different mitovirus species so far defined are less than 40%. (ICTV).

We agree that the previous *Solidago narnavirus 1* is a novel strain of *Fusarium globosum mitovirus 1* based upon ICTV demarcation criteria for the Narnaviridae viral family. In Extended Data Table 5 and elsewhere, this virus is now referred to *Fusarium globosum mitovirus 1* (novel strain 1). This change has also been highlighted in Extended Data Table 5.

P7L150: The "novel" member of Bromoviridae family is most likely a strain of a well-known PNRSV (GeneBank IDs L38823.1, KT444702.1; JN416774.1, extended Data Table 6). Although specific levels of sequence similarity for ilarviruses have not been defined, for species of other genera in this family it is below 80%. 84-93% nucleotide identity most certainly indicate a different strain of PNRSV, rather than a novel virus.

We agree that the previous *Packera aurea bromovirus 1* is a novel strain of *Prunus necrotic ringspot virus* based upon ICTV demarcation criteria for the Bromoviridae viral family. In Extended Data Table 5 and elsewhere, this virus is now referred to *Prunus necrotic ringspot virus* (novel strain 1). This change has also been highlighted in Extended Data Table 5.

P7L151: the same is true for the "novel" species belonging to Secoviridae. Coverage (99%) and percent identity (91-92%) indicate that this could be a strain of Tobacco ringspot nepovirus. Species Demarcation criteria for Secoviridae in conserved Pro-Pol region, acc. to ICTV, are less than 80% aa identity.

We agree that the previous *Oenothera biennis* secovirus 1 is a novel strain of *Tobacco ringspot virus* based upon ICTV demarcation criteria for the Secoviridae viral family. In Extended Data Table 5 and elsewhere, this virus is now referred to *Tobacco ringspot virus* (novel strain 1). This change has also been highlighted in Extended Data Table 5.

P7L149: Importantly, that claims of novelty for these and other viruses in the Extended Table 6 could not be verified independently because sequences are not available/supplied with the manuscript and the Bioproject number PRJNA589022 is not publicly available.

Please see our response to your last comment below regarding Bioproject number PRJNA589022 and availability of sequences from the current study.

P7L159: many viruses in the Supplementary Table 1 could be known species as well, based on the query cover, percent identity and BLAST hits. Novelty of each of those viruses would have to be determined strictly in line with the ICTV criteria for the respective genera.

Using the above family-specific ICTV demarcation criteria, we re-evaluated the status of all the novel partial genomes listed in Supplementary Table 1. With these new criteria, some are now designated as a novel strain of a known virus, but remain in the table. In addition, we added the novel partial strains of the known viruses that were originally listed in the previous Extended Data Table 5 to Supplementary Table 1, for the same reasons. All revisions and additions have been highlighted in Supplementary Table 1.

P8L180-181: members of these three families are long known to be transmitted vertically via seeds and pollen, meaning one does not need a randomization test to reveal that fact.

We wanted to be able to confirm our observations with statistics. Therefore, we think that we should still report the results of the randomization test, especially since not all viruses belonging to these three families may be vertically transmitted via seeds and pollen, and since we discovered novel viruses and strains representing numerous other viral families.

P9L188-189: distribution of viruses found in pollen is likely to be related to their host range. I have not noticed that authors specifically mentioned and/or discussed this important matter.

We think that you raise an excellent point. We have now mentioned this possibility twice in the Discussion, in Lines 287 – 291 and in Lines 327 – 329. The latter builds upon a hypothesis in the sentence immediately before.

P9L201: this is most likely due to airborne bioparticles and fungal spores that are caught in spiky pollen surface patterns. It is however mentioned later in the text (P12L255).

This is an interesting point. We have added a sentence to the Discussion (Lines 317 – 318) as another reason why plants with spiky pollen grains may have more pollen-associated viruses than those with smoother pollen grains.

P10L207: once again, viral richness has to be compared to a host range and occurrence of the respective host plants in those selected geographic areas.

We agree that host range likely plays an important role in structuring the pollen virome, but the full wild plant host range for many plant viruses, including pollen-associated ones, is not known. Characterizing the pollen virome of wild plant species is the first step in addressing this gap in plant virus ecology so that future studies can explore more explicit hypotheses concerning host range, host presence, landscape attributes, and pollen virome richness.

P10L214: I would not call it intriguing: it is quite obvious that pollen grains of species that are in close proximity to anthropogenic environment would carry more contaminants potentially incorporating viral pathogens.

This has been noted, and we have removed the word “intriguing” from the sentence in question.

P10L217: Discussion. Based on the above comments, the discussion section would have to be completely restructured and rewritten.

We agree, thank you for your comments. The Discussion has been greatly revised.

P20L446: please define asymptomatic plants.

Please see our above response to your comment concerning P4, L82 in the Introduction.

P20L449: if only perennial species were sampled, maybe the title of the paper should have been different.

While most of the surveyed plant species were perennials, three were not. Thus, we do not think that we should further revise our title from “Land use and floral traits shape the pollen virome of wild plants”, following one of your earlier suggestions. We have also revised the sentence in question (Lines 528 – 529) to make this more clear.

P20L453: it is unlikely that all debris left after dismembrator could be removed by forceps. Therefore other sources of the described viruses cannot be excluded.

You are correct. We removed visible anther/plant debris and thrips with sterile forceps, and though it is possible that microscopic debris remained, we did not detect common

environmental contaminants in any of our samples. We noted this in the Discussion in response to Reviewer 1 (Lines 338 – 344), and we now include Supplementary Methods files wherein we detail the evaluations of pollen sample purity that we performed.

In the Methods, we have revised the next sentence after the one in question (Lines 536 – 539).

P28L623L: the bioproject PRJNA589022 is not publicly available to verify and evaluate all the sequences reported in the paper.

The raw reads are set to be publicly available in GenBank under Bioproject number PRJNA589022 upon publication of the current study. In the meantime, the contig sequences are available in Supplementary Dataset 2.

This has been made more clear in our “Data availability” statement (Lines 724 – 730).

Reviewer comments, initial review -

Reviewer #1 (Remarks to the Author):

The authors have seriously addressed both major points that I have initially raised, i.e. the pollen purity and the role of fragmentation by agriculture and urbanization on the diversity and distribution of the pollen virome.

This is now a valuable and interesting manuscript.

Reviewer #2 (Remarks to the Author):

I am largely satisfied with the authors' revisions and congratulate them on this interesting study.

I am still not convinced that human disturbance level (rather than any other correlated landscape-level factor that varies among the 4 bioregions) is associated with the pollen virome. The authors suggest that including plant species as replicates in the model can control for this. However, the number of bioregions/levels of disturbance is still 4 no matter how many covariates are included in the model or how many species are replicated in each bioregion. I think that the authors' revised treatment of the results in the discussion is appropriate, but I would suggest that the authors temper the claim in the abstract (L 36-39).

Related: I see that the individual coordinates for each plant species collection are contained in Extended Data Table 3. Although the new land use metrics are calculated for each plant species (lines 256-259) at a 1 and 3 km radius, they are then averaged within a site. If data from within each bioregion can examine the relationship between disturbance and viral diversity, that would significantly enhance the authors' ability to address this relationship in the manuscript.

Other minor comments:

Figure 2: Can the points for each species be added to Fig 2d so that variation among species within each site can be visualized? Similarly, can bioregion be added somehow to Fig 2a-c (perhaps as a point symbol?) The authors did use examine the correlation among predictors (line 702) but did not report a threshold or if trait data varied among sites, so it is unclear if the PC axes are correlated with bioregion.

Was there a phylogenetic signal in the composition or diversity of viral communities among plant species? It appears that the authors have these data (634-636) but I did not see this

statistic reported.

From a look at Extended Data Tables 4-5, it appears that recovery of known viral contigs/genomes was higher in EDAFI (19/23 from EDAFI) while more novel viral genomes were discovered from CC or CA sites (5/7 from CC or CA). Does this suggest that some bioregions or plant species are less investigated for viral diversity than others? Perhaps this is obvious, already discussed or not appropriate for this manuscript.

Reviewer #3 (Remarks to the Author):

Much like after reading the initially submitted manuscript, I remain unconvinced regarding the connection between presence of plant viruses in/on pollen grains and landscape, “human-modified environments” and the flower traits. Successful pollen transmission or for that matter association of plant viruses with pollen, depends primarily on virus species or strain of the virus, type of the host, virus-host interactions, and timing of infection. It also depends on a vector and abiotic factors (wind, water) - in terms of their support for pollination. It is long known from the literature that these factors are more important than the environmental components. I noticed the same doubts and/or suggestions in the comments supplied by other reviewers. Once more, I think that researching exclusively pollen virome, (even on the limited scale of wild species), rather than studying its relationship with land use or environment, would make this study more appropriate for the journal. At its present form and composition, this research is more suitable for specialized, field-specific journals.

Please see the detailed comments below.

P1: the title. I respectfully disagree with this focus statement, which has not been entirely proven in the study.

P2L28-29: these factors may or may not play a role in pollen transmission of viruses. The way this sentence is written assumes that it is a known fact.

P2L32: perhaps, “type of land uses” would be more appropriate

P2L33: “many novel viruses, including three new strains of known viruses”. New strains of known viruses are not novel viral species.

P3L36-39: this seems like a rather vague statement to me. Although there are many types of human modifications of the environment, for the purpose of this research, the agricultural land use appears as the most important.

P2L43-44: successful pollination is a well-known driver of plant-virus interactions.

P3L59-60: Virome is the assemblage of viruses rather than individual pollen-associated viruses. I am not aware of any large-scale, HTS-based studies on pollen virome of agricultural plants. Besides, land use includes agricultural land that is, farming and pastureland. Meaning

that including pollen virome of agricultural crops would certainly make sense.

P4L86-87: pure pollen samples would be required to confirm the host range.

P4L92: my impression was that evaluation of pollen virome was one of the main objectives of this study.

P7L144-146: this is another unclear statement. ICTV species demarcation criteria are different for each group of viruses. Strains are defined not only based on their genetic differences but also on the unique phenotypic characteristics. Viruses whose genomic sequences are different from a reference virus are variants, or isolates.

P7L156: since Fig. 3 is somewhat confusing in terms of definition of novel species or strains, it would be useful to name these eight viruses in the body of the text

P7L158: mostly infect fungi.

P7L158: some tomosviruses are weekly seed transmitted and thus could be pollen transmitted as well.

P7L158: Not exactly true. Some tymoviruses are weakly seed-transmissible (seed-transmitted viruses are often pollen-transmitted) and are also spread by beetles.

P8L71-72: genetic variants or isolates. If this section is devoted only to novel viruses, isolates should be discussed elsewhere.

P8L172: Supplementary information is confusing. I was not able to find the Table named "Extended Data Table 5" with viral sequences in the suppl. information to confirm the authors' claims. Here is the list of all supplementary files:

Supplementary Information Guide PDF (14KB)

Source File (PDF) 14KB

Supplementary Table 1 PDF (360KB) Source File (PDF) 360KB

Supplementary Methods PDF (99KB) Source File (DOCX) 32KB

Supplementary Methods Table 1 PDF (7KB) Source File (DOCX) 13KB

Supplementary Methods Table 2 PDF (152KB) Source File (DOCX) 13KB

Supplementary Methods Table 3 PDF (80KB) Source File (DOCX) 18KB

Supplementary Methods Figure 1 PDF (223KB) Source File (PDF) 223KB

Supplementary Dataset 1 Supplementary Dataset (72KB) Source File (XLSX) 72KB

Supplementary Dataset 2 Supplementary Dataset (407KB) Source File (XLSX) 407KB

Source Data Supplementary Dataset (19KB) Source File (XLSX) 19KB

According to the Supplementary Information guide, all viral sequences are in the Supplementary Dataset 2. Judging by the sequence lengths, I could not find any complete genomes there. Also, the viral tentative names appear to be different from those in Fig.3. The list of supplementary information in the manuscript differs from the one in the manuscript tracking system.

P8L177: there are five genera in the family Partitiviridae. Where do these two novel partitiviruses belong? Alphapartivirus or Betapartivirus?

P10L208: "Viral family and host plant subclass determine pollen-associated virus

distributions". This would have to be rephrased. Family and subclass are taxonomic ranks. Taxonomic ranks cannot determine virus distribution. It is defined by the biology of virus-host interactions.

P10L215: once again, the mechanisms of pollen transmission of these viruses are known.

P13L284-285: this could be directly related to the host range of the viruses rather than to the land use patterns

P14L306-307: I do not think that this relationship was convincingly established in this research.

P23-24L524-526: if pollen-associated viruses were the main topic of this research, it is not clear why plants with typical virus symptomatology were not sampled?

Dear Reviewers,

Thank you for your comments. We have fully addressed all of them, including the main suggestion of tempering the conclusions about the environmental components that influence the pollen virome. The paper is even clearer and stronger than our second resubmission. Please find below a summary of the changes made in response to your comments (in **bold**). The line numbers refer to the track-changes version of the manuscript that we have uploaded for the current resubmission. We have also uploaded new Extended Data and Supplementary Information files for the current resubmission.

Sincerely,
Corresponding authors, for all authors.

Responses to Reviewer #1

The authors have seriously addressed both major points that I have initially raised, i.e., the pollen purity and the role of fragmentation by agriculture and urbanization on the diversity and distribution of the pollen virome.

This is now a valuable and interesting manuscript.

Thank you!

Responses to Reviewer #2

I am largely satisfied with the authors' revisions and congratulate them on this interesting study.

Thank you!

I am still not convinced that human disturbance level (rather than any other correlated landscape-level factor that varies among the 4 bioregions) is associated with the pollen virome. The authors suggest that including plant species as replicates in the model can control for this. However, the number of bioregions/levels of disturbance is still 4 no matter how many covariates are included in the model or how many species are replicated in each bioregion. I think that the authors' revised treatment of the results in the discussion is appropriate, but I would suggest that the authors temper the claim in the abstract (L 36-39).

We agree that other landscape-level factors may covary with human land use among the four regions, so we have tempered the sentence accordingly (Lines 37 – 40 in the Abstract): “Across the four regions, wild plant species harbored more viruses when surrounded by less natural vegetation and more human-modified environments (e.g., agriculture) than the reverse, but we note that other region-level differences may also covary with pollen-associated virus richness and should be explored to pinpoint the drivers of this pattern.”.

Related: I see that the individual coordinates for each plant species collection are contained in Extended Data Table 3. Although the new land use metrics are calculated for each plant species

(lines 256-259) at a 1 and 3 km radius, they are then averaged within a site. If data from within each bioregion can examine the relationship between disturbance and viral diversity, that would significantly enhance the authors' ability to address this relationship in the manuscript.

While we agree that land use data for each plant species within each region could be useful, for the scale at which we collected pollen, the variance in land use within the regions was very small (ranging from 0 to 0.025, Lines 142 – 143) and therefore did not provide much discriminatory power. Thus, we did not include this level of analysis.

Other minor comments:

Figure 2: Can the points for each species be added to Fig 2d so that variation among species within each site can be visualized? Similarly, can bioregion be added somehow to Fig 2a-c (perhaps as a point symbol?)

Thank you for your suggestion for Fig. 2d. We have added the points for each plant species. Similarly, we have also added points for each plant species to Extended Data Figure 2b.

We did not add region indicators to Fig. 2a – c (and Extended Data Figure 2a), however, as it would make the panels difficult to read. Nevertheless, the information regarding in which region a plant species was sampled is presented in Figs. 1 and 3 and Extended Data Tables 1 and 3.

The authors did use examine the correlation among predictors (line 702) but did not report a threshold or if trait data varied among sites, so it is unclear if the PC axes are correlated with bioregion.

We performed analyses of variance (ANOVAs) to examine whether the PCs varied between the regions and found that PC1 significantly varied between the regions (df = 3, F = 3.23, $P = 0.04$). Post-hoc Tukey's test revealed only modest differences between the California Grassland and California Coast regions ($P = 0.08$) and the Central Appalachian and California Grasslands regions ($P = 0.08$). There were no meaningful differences in PC1 between any other pairs of regions (all $P > 0.36$). PCs 2 and 3 did not vary significantly between the regions (df = 3, F = 1.59, $P = 0.22$ and df = 3, F = 1.93, $P = 0.16$, respectively). We have added this information to Lines 125 – 132.

Was there a phylogenetic signal in the composition or diversity of viral communities among plant species? It appears that the authors have these data (634-636) but I did not see this statistic reported.

There were phylogenetic signals in the estimates of virus richness (i.e., diversity) among the plant species. This method is reported in Lines 648 – 651 and the accompanying statistics are reported in Lines 228 – 230: “Neither the conservative nor relaxed virus richness estimates were significantly influenced by plant evolutionary history (Pagel's $\lambda = 0.35, 0.42$, respectively; $P = 0.34, 0.31$, respectively).”

From a look at Extended Data Tables 4-5, it appears that recovery of known viral contigs/genomes was higher in EDAFI (19/23 from EDAFI) while more novel viral genomes were discovered from CC or CA sites (5/7 from CC or CA). Does this suggest that some bioregions or plant species are less investigated for viral diversity than others? Perhaps this is obvious, already discussed or not appropriate for this manuscript.

You raise a very interesting point! Most known plant viruses were discovered in agricultural species, so it stands to reason that we found more known viruses near agriculture. We have added a few sentences to our Discussion (Lines 282 – 293) to highlight this possibility: “Interestingly, we also found more known viruses in association with pollen from the Eastern Deciduous Agro-forest Interface (Extended Data Table 4) and many novel viruses in association with pollen from the California Coast and Central Appalachian regions (Extended Data Table 5; Supplementary Table 1). This pattern may reflect the facts that many known plant viruses were identified from agricultural crops rather than wild species and that the plants in the Eastern Deciduous Agro-forest Interface are surrounded by more agricultural land use than the plants in the California Coast and Central Appalachian regions.”.

Responses to Reviewer #3

Much like after reading the initially submitted manuscript, I remain unconvinced regarding the connection between presence of plant viruses in/on pollen grains and landscape, “human-modified environments” and the flower traits. Successful pollen transmission or for that matter association of plant viruses with pollen, depends primarily on virus species or strain of the virus, type of the host, virus-host interactions, and timing of infection. It also depends on a vector and abiotic factors (wind, water) - in terms of their support for pollination. It is long known from the literature that these factors are more important than the environmental components. I noticed the same doubts and/or suggestions in the comments supplied by other reviewers. Once more, I think that researching exclusively pollen virome, (even on the limited scale of wild species), rather than studying its relationship with land use or environment, would make this study more appropriate for the journal. At its present form and composition, this research is more suitable for specialized, field-specific journals.

We have carefully considered your thoughtful comments and made the following changes:

- 1. As we have noted in our response to Reviewer #2, we tempered our conclusions regarding the direct effect of human land use on the pollen virome in several places (see above; including Lines 37 – 40).**
- 2. We revised our terminology to be more precise regarding transmission of pollen-associated viruses via pollinators. Specifically, we now differentiate between *pollen-associated virus transmission* and *pollen transport*. The former refers to the infection of individual plants by viruses that are pollen-associated, and the latter refers to the transport of pollen-associated viruses to plants via pollen carried on the pollinator vector.**

These terms have been clarified in the following lines:

Lines 49 – 51: “The association of viral diversity with floral traits highlights the need to incorporate plant-pollinator interactions as a driver of pollen-associated virus transport into the study of plant-viral interactions.”

Lines 102 – 104: “...whether pollen-associated virus taxonomic richness correlated with floral and pollen grain traits important for plant interactions with pollinators that serve as pollen vectors.”

Lines 308 – 311: “Our study uncovered a previously unrecognized relationship between pollen virome richness and plant traits that attract pollinators and influence their interactions with pollen, thus by extension highlighting the potential importance of pollinators as vectors of pollen-associated viruses.”

Lines 323 – 325: “Likewise, since smaller pollen grains are potentially easier for pollinators to handle and pack into pollen loads^{e.g.,55-56}, pollinators as vectors of pollen-associated viruses may preferentially visit plant species that produce smaller grains.”

Lines 350 – 354: “The identification of several plant traits that increase the association between plants and their pollinators, who are important pollen-associated virus vectors, as well as the land use patterns correlated with virus richness, expand our knowledge of viral host ranges and recognize for the first time the diversity of viruses that could be pollinator-transmitted.”

3. The main goals of our study were to first describe the pollen virome of wild plant species and then to shed the first light on the plant and landscape factors that are associated with its diversity. We have revised the sentences that list the goals of the paper in the Introduction to provide a more balanced view of the objectives and weight of the results:

Lines 93 – 104: “We leveraged the power of metagenomics and wide species-level sampling to characterize the pollen viromes of 24 wild, visually asymptomatic plant species (from 16 families and five subclasses), each growing in one of four geographic regions in the United States (Fig. 1). We then used phylogenetically controlled analyses to address the following: 1) whether pollen-associated viruses were limited to a few, previously recognized viral families; 2) whether pollen-associated viruses were heterogeneously distributed across geographic regions differing in amounts of human land use and across plant subclasses; and 3) whether pollen-associated virus taxonomic richness correlated with floral and pollen grain traits important for plant interactions with pollinators that serve as pollen vectors.”

Please see the detailed comments below.

P1: the title. I respectfully disagree with this focus statement, which has not been entirely proven in the study.

We have changed the title to “The pollen virome of wild plants and its association with variation in floral traits and land use” to better reflect the foci of our study.

P2L28-29: these factors may or may not play a role in pollen transmission of viruses. The way this sentence is written assumes that it is a known fact.

We have tempered this sentence (Lines 28 – 32), and it now reads: “To discover the diversity of pollen-associated viruses and uncover landscape and floral features that may contribute to it, we performed a species-level metagenomic survey of pollen from wild, visually asymptomatic plants (24 species, 16 families, five subclasses), located in one of four regions in the United States that vary in types of land use.”.

P2L32: perhaps, “type of land uses” would be more appropriate

The sentence in question (Lines 28 – 32) now ends with “...types of land use.”.

P2L33: “many novel viruses, including three new strains of known viruses”. New strains of known viruses are not novel viral species.

The sentence in question (Lines 32 – 34) has been edited to be more clear: “We identified 22 known viruses and many novel viruses, including six coding-complete viral genomes and three coding-complete variants of known viruses, as pollen-associated.”.

We have also replaced the term “strain” with the term “variant” throughout the manuscript text, figures, extended data, and supplementary information (see below).

P3L36-39: this seems like a rather vague statement to me. Although there are many types of human modifications of the environment, for the purpose of this research, the agricultural land use appears as the most important.

We have tempered our conclusions in the sentence in question (Lines 37 – 40) in response to Reviewer #2 and have also indicated that agriculture is an example of “human-modified environments”: Across the four regions, wild plant species harbored more viruses when surrounded by less natural vegetation and more human-modified environments (e.g., agriculture) than the reverse, but we note that other region-level differences may also covary with pollen-associated virus richness and should be explored to pinpoint the drivers of this pattern.”.

P2L43-44: successful pollination is a well-known driver of plant-virus interactions.

As mentioned above, we now use more precise language to differentiate between *pollen transmission* and *pollen transport* of viruses, thus we have changed the last two sentences of the Abstract (Lines 45 – 51) to more clearly show that our focus is on floral traits important for the pollen transport stage of the plant-pollen-associated virus interaction:

“When examining the novel connection between virus richness and traits related to the plant-pollinator interaction that affect pollen transport, we found that plant species with multiple, bilaterally symmetric flowers and smaller, spikier pollen harbored more viruses

than those with single, radially symmetric flowers and larger, smoother pollen grains. The association of viral diversity with floral traits highlights the need to incorporate plant-pollinator interactions as a driver of pollen-associated virus transport into the study of plant-viral interactions.”

P3L59-60: Virome is the assemblage of viruses rather than individual pollen-associated viruses. I am not aware of any large-scale, HTS-based studies on pollen virome of agricultural plants. Besides, land use includes agricultural land that is, farming and pastureland. Meaning that including pollen virome of agricultural crops would certainly make sense.

We have changed the sentence in question to clarify the nature of past studies on pollen-associated viruses since ours is the first high-throughput study on the pollen virome of either wild or agricultural plant species (Lines 66 – 68): “Still, our knowledge of the pollen virome as a whole is sparse and weighted toward mechanistic studies on viral infection using agricultural plant species.”.

We agree that future high-throughput studies on the pollen virome should include agricultural crops.

P4L86-87: pure pollen samples would be required to confirm the host range.

The sentence (Lines 90 – 93) now reads: “Our metagenomics approach allowed us to capture all viruses present^{e.g.,40-45}, including pathogenic, neutral, and possibly mutualistic ones³⁸, as well as to identify known viruses in hosts not previously recognized to *perhaps* be within their host ranges⁴⁶ and novel viruses not previously detected or described.”.

P4L92: my impression was that evaluation of pollen virome was one of the main objectives of this study.

It was, you are correct. We have edited the last paragraph of the Introduction to make this point stand out:

Lines 88 – 90: “To address fundamental gaps in the knowledge of pollen-associated plant viruses, our first goal was to undertake a metagenomic study of the pollen virome.”

Lines 93 – 104: “We leveraged the power of metagenomics and wide species-level sampling to characterize the pollen viromes of 24 wild, visually asymptomatic plant species (from 16 families and five subclasses), each growing in one of four geographic regions in the United States (Fig. 1). We then used phylogenetically controlled analyses to address the following: 1) whether pollen-associated viruses were limited to a few, previously recognized viral families; 2) whether pollen-associated viruses were heterogeneously distributed across geographic regions differing in amounts of human land use and across plant subclasses; and 3) whether pollen-associated virus taxonomic richness correlated with floral and pollen grain traits important for plant interactions with pollinators that serve as pollen vectors.”

P7L144-146: this is another unclear statement. ICTV species demarcation criteria are different

for each group of viruses. Strains are defined not only based on their genetic differences but also on the unique phenotypic characteristics. Viruses whose genomic sequences are different from a reference virus are variants, or isolates.

It is true that the ICTV family-specific species demarcation criteria encompasses both genetic and phenotypic differences. Therefore, we have clarified the fact that we used only family-specific percent identity species demarcation thresholds as our initial designation of “known” vs “novel” (Lines 150 – 155).

At your suggestion and because of the following definitions from Kuhn *et al.* (2013), we have also changed every use of “strain” to “variant” throughout the manuscript and all associated files. However, there may not be universally accepted definitions of strain, variant, and isolate in virology (Kuhn *et al.*, 2013).

Strain: “According to Van Regenmortel (2007), a (natural) virus strain is a ‘variant of a given virus that is recognizable because it possesses some unique phenotypic characteristics that remain stable under natural conditions’...”.

Variant: “Van Regenmortel (2007) defined a virus variant as ‘an isolate or a set of isolates whose genomic (consensus) sequence(s) differ(s) from that of a reference virus’...”.

Isolate: “Fauquet and Stanley (2005) defined a virus isolate as ‘a sample...that has been cultured for study.’ Van Regenmortel (2007) has come to a similar conclusion and defined a virus isolate as ‘simply an instance of a particular virus’...”.

References

Fauquet CM, Stanley J. 2005. Revising the way we conceive and name viruses below the species level; a review of geminivirus taxonomy calls for new standardized isolate descriptors. *Archives of Virology* 150: 2151 – 2179.

Van Regenmortel MHV. 2007. Virus species and virus identification: past and current controversies. *Infection, Genetics, and Evolution* 7: 133 – 144.

Kuhn JH, Bao Y, Bavari S, Becker S, Bradfute S, Brister JR *et al.* 2013. Virus nomenclature below the species level: a standardized nomenclature for natural variants of viruses assigned to the family *Filoviridae*. *Archives of Virology* 158: 301 – 311.

P7L156: since Fig. 3 is somewhat confusing in terms of definition of novel species or strains, it would be useful to name these eight viruses in the body of the text

We have added the virus names to the sentence in question (Lines 160 – 164): “All but four of these (*Deformed wing virus*, *Alternaria arborescens mitovirus 1*, *Fusarium globosum mitovirus 1*, and *Hubei narna-like virus 25*) are classified as plant viruses, and only eight (*Apple mosaic*, *Prunus necrotic ringspot*, *Strawberry necrotic shock*, *Tobacco streak*, *Cherry rasp leaf*, *Tobacco ringspot*, *Tomato ringspot*, and *Deformed wing viruses*) have been previously described as being pollen-associated.”.

P7L158: mostly infect fungi.

Yes! At the end of the paragraph in question, we already state that the two known viruses belonging to the Narnaviridae family infect fungi (Lines 172 – 173): “Two of the other non-plant viruses detected, *Alternaria arborescens mitovirus 1* and *Fusarium globosum mitovirus 1*, infect fungi⁴⁸.”.

P7L158: some tombusviruses are weekly seed transmitted and thus could be pollen transmitted as well.

You are correct. We have edited the sentence to include these points (Lines 165 – 169) as follows: “These include members of the Narnaviridae, Tombusviridae, and Tymoviridae, three viral families with no or little previously documented association with pollen, although some tombusviruses and tymoviruses are weakly seed-transmitted and thus could be pollen-transmitted as well (Extended Data Table 4).”.

P7L158: Not exactly true. Some tymoviruses are weakly seed-transmissible (seed-transmitted viruses are often pollen-transmitted) and are also spread by beetles.

Yes, addressed in the preceding comment.

P8L71-72: genetic variants or isolates. If this section is devoted only to novel viruses, isolates should be discussed elsewhere.

We respectfully disagree. Though the three coding-complete variants are not new viral species, they are still previously undiscovered variants. They are now noted as ‘variants’ throughout the manuscript and all manuscript-associated files to clarify this point.

P8L172: Supplementary information is confusing. I was not able to find the Table named “Extended Data Table 5” with viral sequences in the suppl. information to confirm the authors' claims. Here is the list of all supplementary files:

Supplementary Information Guide PDF (14KB)

Source File (PDF) 14KB

Supplementary Table 1 PDF (360KB) Source File (PDF) 360KB

Supplementary Methods PDF (99KB) Source File (DOCX) 32KB

Supplementary Methods Table 1 PDF (7KB) Source File (DOCX) 13KB

Supplementary Methods Table 2 PDF (152KB) Source File (DOCX) 13KB

Supplementary Methods Table 3 PDF (80KB) Source File (DOCX) 18KB

Supplementary Methods Figure 1 PDF (223KB) Source File (PDF) 223KB

Supplementary Dataset 1 Supplementary Dataset (72KB) Source File (XLSX) 72KB

Supplementary Dataset 2 Supplementary Dataset (407KB) Source File (XLSX) 407KB

Source Data Supplementary Dataset (19KB) Source File (XLSX) 19KB

Extended Data Table 5 is not included in the Supplementary Information. It is instead included in the Extended Data file, which, following journal submission formatting,

includes all six Extended Data Tables and both Extended Data Figures. The Extended Data file has been included in each submission of the manuscript, including the current resubmission.

All contig/extended contig sequences are included in Supplementary Dataset 2, not Extended Data Table 5. Extended Data Table 5 is derived from Supplementary Dataset 2 and lists only the six novel coding-complete viral genomes and the three novel coding-complete variants of known viruses that we discovered in our pollen samples.

According to the Supplementary Information guide, all viral sequences are in the Supplementary Dataset 2. Judging by the sequence lengths, I could not find any complete genomes there. Also, the viral tentative names appear to be different from those in Fig.3. The list of supplementary information in the manuscript differs from the one in the manuscript tracking system.

All contig/extended contig sequences are included in Supplementary Dataset 2. We have made an effort to better connect Figure 3, Extended Data Table 5, Extended Data Figure 1, and Supplementary Table 1 to the information in Supplementary Dataset 2. Doing so should have resolved any discrepancies that may have been present in viral names between the files in previous submissions.

Our review of the lists of Supplementary Information in the manuscript and the manuscript tracking system reveal no differences between them. The Supplementary Information files included in previous submissions and the current one are: Supplementary Information Guide, Supplementary Table 1, Supplementary Methods, Supplementary Methods Tables 1 – 3, Supplementary Methods Figure 1, and Supplementary Datasets 1 – 2. Outside of the Supplementary Information is the Source Data file for all the figures and the Extended Data file, which includes all six Extended Data Tables and both Extended Data figures.

P8L177: there are five genera in the family Partitiviridae. Where do these two novel partitiviruses belong? Alphapartivirus or Betapartivirus?

The top RapSearch2 hit in the NCBI protein and nucleotide databases to *Ranunculus californicus* partitivirus 1 is *Rose partitivirus* (Supplementary Dataset 2), which may be in the *Alphapartivirus* genus (ICTV, 2021), though it is currently listed as an unclassified Partitiviridae in the NCBI Taxonomy database. Due to this uncertainty, we do not classify *Ranunculus californicus* partitivirus 1 into a genus.

The top RapSearch2 hits in the NCBI protein and nucleotide databases to *Packera aurea* partitivirus 1 are *Raphanus sativus cryptic virus 3* and *Sinapis alba cryptic virus 1* (Supplementary Dataset 2). *Raphanus sativus cryptic virus 3* may be in the *Alphacryptovirus* genus (ICTV, 2021), though it is currently listed as an unclassified Partitiviridae in the NCBI Taxonomy database. *Sinapis alba cryptic virus 1* is currently listed as an unclassified Partitiviridae in the NCBI Taxonomy database. Due to this uncertainty, we do not classify *Packera aurea* partitivirus 1 in a genus.

P10L208: “Viral family and host plant subclass determine pollen-associated virus distributions”. This would have to be rephrased. Family and subclass are taxonomic ranks. Taxonomic ranks cannot determine virus distribution. It is defined by the biology of virus-host interactions.

To better match the language in the last paragraph of the Introduction, we have changed this heading to “*Pollen-associated viruses are heterogeneously distributed across viral families and plant subclasses*” (Lines 213 – 214).

P10L215: once again, the mechanisms of pollen transmission of these viruses are known.

We have edited the sentence to acknowledge the previous work showing that members of the Bromoviridae, Partitiviridae, and Secoviridae viral families have known mechanisms of pollen transmission.

This sentence (Lines 217 – 221) now states: “A permutation test revealed that this distribution of viruses is significantly different from random chance, providing confirmatory evidence that these viral families have members with characteristics that allow for the exploitation of the pollen niche (observed = 39, 95% CI = 27 – 38, $P < 0.05$).”.

P13L284-285: this could be directly related to the host range of the viruses rather than to the land use patterns

We now acknowledge this possibility and have edited this paragraph in response to Reviewer #2 and also Lines 296 – 300 now read: “In fact, if the viral diversity–land use patterns seen here are due to human disturbance and the potential for viral spillover rather than reflecting viral host range, then we would predict significant variation in pollen-associated virus incidence or diversity within and among populations, dependent upon proximity to agriculture or other human-disturbed habitats.”.

P14L306-307: I do not think that this relationship was convincingly established in this research.

We respectfully disagree, but have edited the sentence to make explicit what we have established in this research. Specifically, our analyses clearly show significant correlations between plant traits important for attracting pollinators, as well as their interactions with flowers and pollen (multiple-flowered inflorescences, bilaterally symmetric flowers, restricted access to floral rewards; smaller, spikier pollen grains) and pollen-associated virus richness.

Thus, we have revised the sentence (Lines 308 – 311) to: “Our study uncovered a previously unrecognized relationship between pollen virome richness and plant traits that attract pollinators and influence their interactions with pollen, thus by extension highlighting the potential importance of pollinators as vectors of pollen-associated viruses.”.

P23-24L524-526: if pollen-associated viruses were the main topic of this research, it is not clear why plants with typical virus symptomatology were not sampled?

We chose to sample plants that were not visibly infected because we did not want to bias pollen-associated virus recovery or identification.

Reviewer comments, further review -

Reviewer #2 (Remarks to the Author):

My comments have been thoroughly addressed. Congrats again to the authors on this interesting and important paper.

Response to Reviewers

Reviewer #2

My comments have been thoroughly addressed. Congrats again to the authors on this interesting and important paper.

Thank you!